# Bond performance between hybrid fiber-reinforced concrete and BFRP bars under freeze-thaw cycle

Yanming Su ◉ *

School of Civil Engineering, Shenyang Jianzhu University, Shenyang, China

* 844240921@qq.com

## Abstract

This study applied the pull-out test to examine the influence of freeze-thaw cycles and hybrid fiber incorporation on the bond performance between BFRP bars and hybrid fiber-reinforced concrete. The bond-slip curves were fitted by the existing bond-slip constitutive model, and then the bond strength was predicted by a BP neural network. The results indicated that the failure mode changed from pull-out to splitting for the BFRP bar ordinary concrete specimens when the freeze-thaw cycles exceeded 50, while only pull-out failure occurred for all BFRP bar hybrid fiber-reinforced concrete specimens. An increasing trend was shown on the peak slip, but a decreasing trend was shown on the bond stiffness and bond strength when freeze-thaw cycles increased. The bond strength could be increased significantly by the incorporation of basalt fiber (BF) and cellulose fiber (CF) under the same freezing and thawing conditions as compared to concrete specimens without fibers. The Malvar model and the Continuous Curve model performed best in fitting the ascending and descending sections of the bond-slip curves, respectively. The BP neural network also accurately predicted the bond strength, with relative errors of predicted bond strengths ranging from 3.75% to 13.7%, and 86% of them being less than 10%.

## Introduction

The reinforced concrete structure is the most common form of a building structure because of its advantages of high integrity, easy casting, and low cost [1–5]. However, the steel bar might corrode when the reinforced concrete structure was in specific environments, such as a seaport wharf or a de-icing salt pavement, etc [6, 7]. The problem was even more severe in most of the frozen soil areas, and could impair the bond performance between steel bars and concrete, the structural bearing capacity, and the structural service life [8–10]. Basalt Fiber Reinforced Polymer (BFRP) bar was a new green and environmentally friendly inorganic material that had been developed and used rapidly because of its high tensile strength and durability [11, 12]. By replacing conventional steel bars with them, the problem of corrosion of steel bars could be avoided.

Besides solving the corrosion problem by replacing steel bars with BFRP bars, the durability of the bond performance between BFRP bars and concrete in various environments also needs to be examined. Scholars had conducted experimental research on different accelerated

**Data Availability Statement:** All relevant data are within the manuscript and its Supporting Information files.

**Funding:** This work was supported by a grant from the National Natural Science Foundation of China

(No. 51938009). the funders had no role in study design, data collection and analysis, decision to publish, or preparation of the manuscript.

**Competing interests:** The authors have declared that no competing interests exist.

environment tests. For high temperature environments, Li et al. [13] studied the bond performance between BFRP bars and concrete and the residual bond performance in the temperature ranging from 20°C to 350°C. They found that the bond strength reduced when temperature increased. When the temperature reached 350°C, the resinous polymer was totally carbonized and the bond strength was severely decreased. The residual bond strength was only 12.2% of that at room temperature at that point. Thus, the high-temperature environment seriously deteriorated the bond strength. For acidic, alkaline, and seawater environments, etc., Altalmas et al. [14] subjected BFRP bar concrete specimens to acidic, alkaline, and seawater environments at 60°C before performing a central pull-out test. They observed that the bond strengths of the BFRP bar concrete specimens after 90 days decreased by 25%, 25%, and 14% in environments of seawater, alkaline, and acidic, respectively. The bond strength under the combined effects of temperature variation and seawater environment was studied by Wu et al. [15] through pull-out tests on BFRP bar concrete specimens immersed in artificial seawater. They observed that the bond strength increased and then decreased with the immersion time at room temperature, which was attributed to the moisture expansion of the bar. However, at higher temperatures (40°C–55°C), the bond strength decreased with the immersion time, as the degradation rate was accelerated by the increase in temperature. Khanfour et al. [16] examined the impact of freeze-thaw cycles on the bond strength between BFRP bars and concrete by subjecting BFRP bar concrete specimens to freeze-thaw cycle tests at temperatures ranging from -25°C to 15°C. They found that the bond strengths and bond-slip curves were not affected more by the freeze-thaw cycles, nor was the failure mode of specimens. However, Wu et al. [17] reported that mortar collapse was shown on the surface of the concrete specimens after 40 freeze-thaw cycles, when they performed a central pull-out test on them. They also noted that the bond strength showed a first increasing and then decreasing trend with the increase of freeze-thaw cycles, which could be explained by the secondary hydration reaction that occurred inside the concrete.

Concrete was also subject to different degrees of deterioration in an erosion environment [18–21]. Previous extensive research found that fiber could enhance the frost resistance of concrete [22–25]. BF, a novel green fiber derived from basalt melting, had high resistance to acids and alkalis, good stability, and a significant role in strengthening the durability and mechanical properties of concrete [26]. Jin et al. [27] studied the effect of BF with different volume content (0%, 0.1%, 0.2%, and 0.3%) on the frost resistance of concrete. They found that the frost resistance improved with the increase of fiber volume content through comparative experimental research. They also reported that 0.3% volume content of BF compacted the concrete and reduced the energy loss during freezing and thawing, which had a beneficial effect on the enhancement of the frost resistance. Yang et al. [28] found that the splitting tensile strength and flexural strength of concrete increased with the increase of BF volume content after freeze-thaw cycles. CF was a new generation of high-performance plant fiber. It has a unique cavity structure and natural hydrophilicity, which can enhance the cracking resistance and the impermeability of concrete, thereby increasing the frost resistance durability of concrete [29]. The effect of CF on the freeze-thaw resistance of concrete was investigated by Li et al. [30], who reported that the incorporation of CF slowed the rate of decline in relative dynamic elastic modulus and increased the maximum number of freeze-thaw cycles that the specimens could withstand. This indicated that cellulose fiber reinforced concrete had improved frost resistance.

The experimental results from previous studies varied widely due to different test conditions and other factors, which reduced the comparability of relevant research conclusions. The bond durability between the BFRP bar and concrete under an erosion environment was still underexplored, as environmental factors were often neglected in the existing studies, which

made it hard to reflect the bond-slip relationship in engineering practice. The purpose of this research was to study the influences of freeze-thaw cycles and the incorporation of hybrid fiber (BF and CF) on the bond performance between BFRP bars and concrete. The bond-slip curves were examined, and then they were fitted and compared with the bond-slip constitutive models. Moreover, the bond strengths were predicted by BP artificial neural network. Notably, a theoretical reference was provided for the practical application of BFRP bars in freeze-thaw environments.

## Experimental program

### Material

The cubic compressive strength was 40MPa in this test, and the concrete mixture proportion was designed according to JGJ 55–2011 [31] which was: $m_{cement}$: $m_{sand}$: $m_{gravel}$: $m_{water}$ = 1: 1.46: 2.07: 0.48. Normal Portland cement (P.O 42.5) produced by Bohai Cement (Huludao) Co., Ltd. was used in this test. The coarse aggregate consisted of natural gravel with a continuous gradation in the particle size range of 5–20 mm, and its gradation curve obtained from the test according to GB/T 14685–2022 [32] is shown in Fig 1. The bulk density, apparent density, mud content, and crushing index of gravel were 1630 kg/m$^3$, 2680 kg/m$^3$, 0.3%, and 8.9%, respectively. The fine aggregate was natural river sand with a fineness modulus of 2.7, and urban tap water supplied by the Jinzhou City Tap Water Plant was used for water.

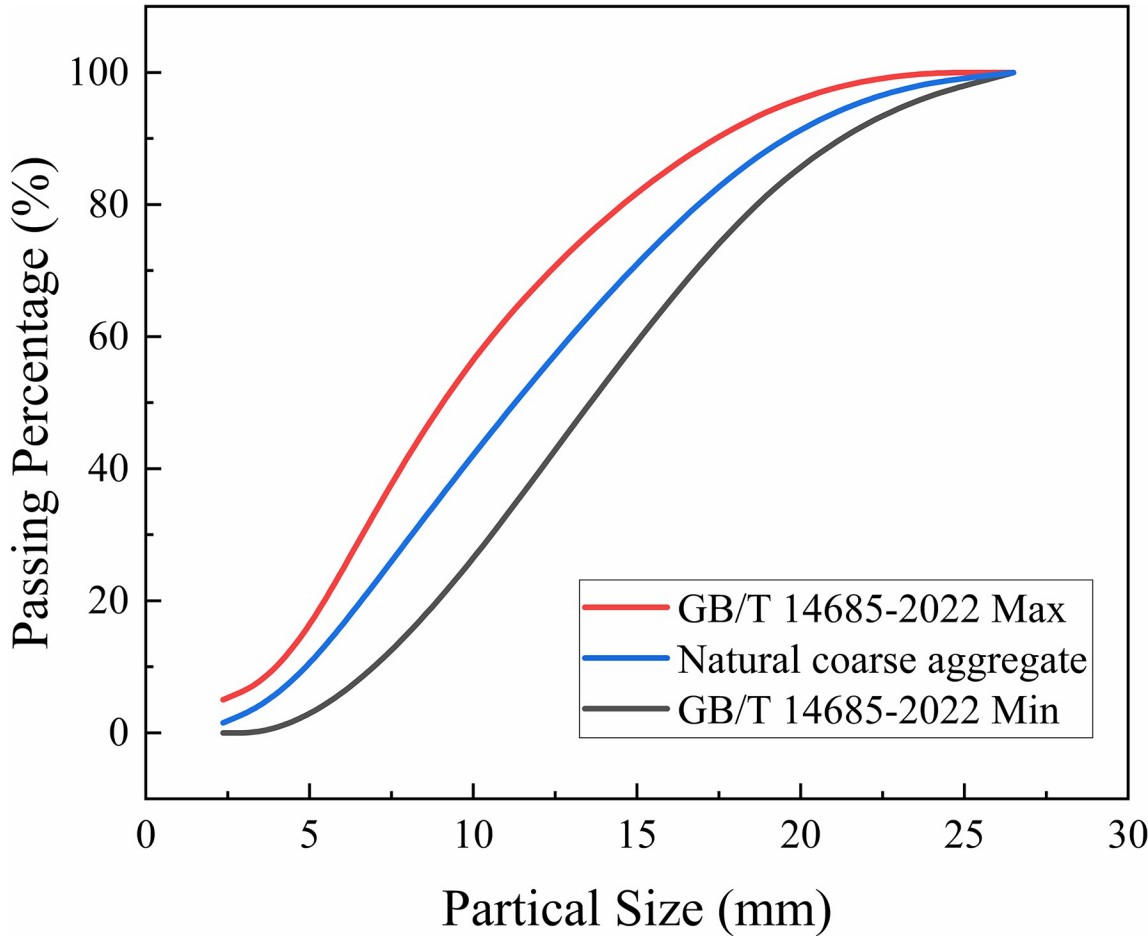

**Fig 1. Gradation curves of coarse aggregate.**

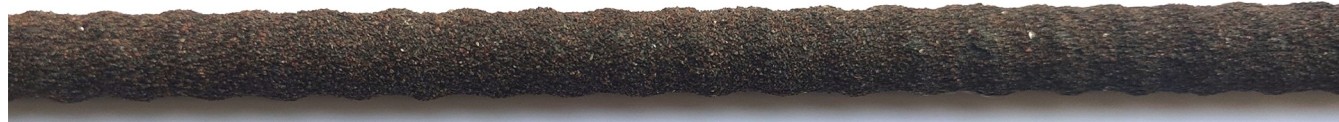

**Fig 2. Appearance morphology of BFRP bar.**

Table 1. Material properties of BFRP bar.

| Cross section area (mm$^2$) | Tensile strength (MPa) | Elastic modulus (GPa) | Density (g/cm$^3$) | Elongation at break (%) | Rib spacing (mm) | Rib height (mm) |
|---|---|---|---|---|---|---|
| 113.10 | 1216 | 52.5 | 2.02 | 2.3 | 10 | 1.4 |

The BFRP bars were produced by Jiangsu Green Material Valley New Material Technology Development Co., Ltd, with a diameter of 12 mm. They had a sand-coated and ribbed surface treatment, and their apparent morphology and material properties are shown in Fig 2 and Table 1, respectively. The BF and CF were produced by Jiangsu Tianlong Continuous Basalt Fibre Co., Ltd. and Changzhou Leade New Materials Co., Ltd., respectively. Their fiber apparent morphology and material properties are shown in Fig 3 and Table 2.

## Design of pull-out specimens

This test focused on the qualitative study, so the effect of various fiber volume fractions was not investigated. The volume content of CF and BF were set as 0.15% and 0.2% of the concrete volume, respectively, and the number of freeze-thaw cycles was set as 0, 25, 50, 75, and 100, based on references [33, 34]. Three specimens from each of the 10 groups of cube center

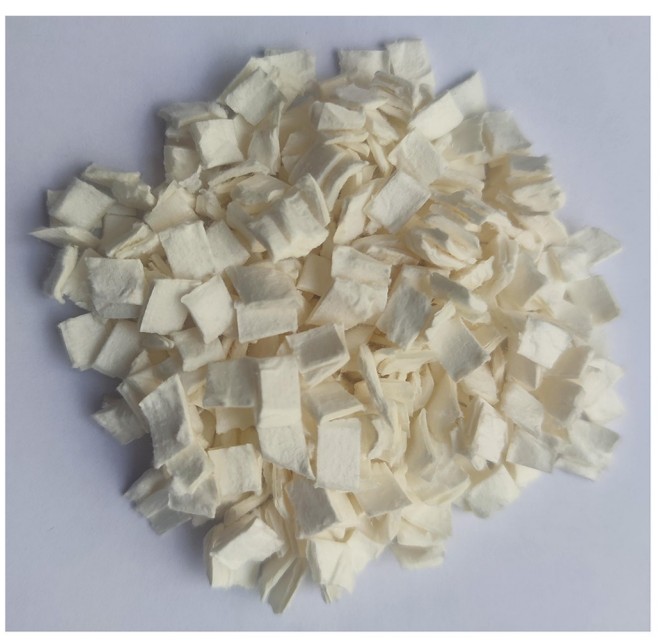
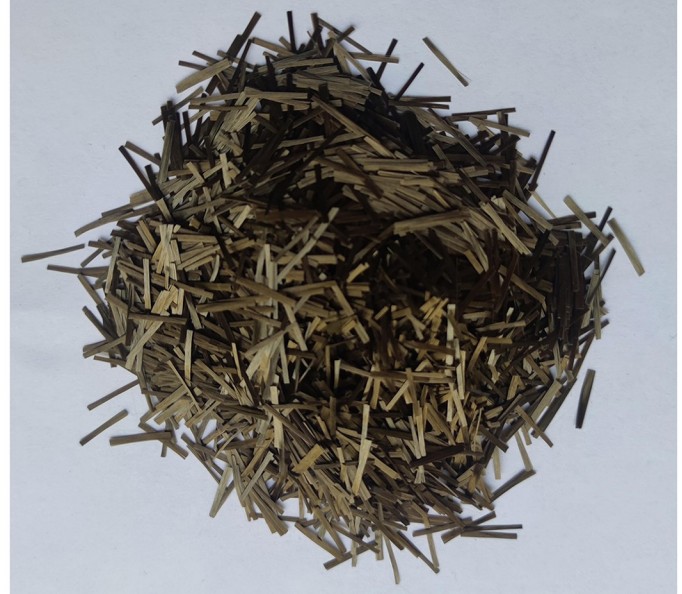

(a) BF apparent morphology (b) CF apparent morphology

**Fig 3. Fiber apparent morphology.**

**Table 2. Material properties of fiber.**

| Fiber type | Length (mm) | Diameter (µm) | Elastic modulus (GPa) | Tensile strength (MPa) | Density (g/cm³) | Elongation at break (%) |
|---|---|---|---|---|---|---|
| BF | 12 | 16 | 90~110 | 3000~3500 | 2.65~3.05 | 3.2 |
| CF | 2.1 | 17.9 | 9.25 | 913 | 1.11 | - |

pullout specimens were cast. For example, FT50-C0.15B0.2 denoted the specimens with 50 freeze-thaw cycles, 0.15% CF volume content, and 0.2% BF volume content. Moreover, according to GB/T50081-2019 [35], the cube specimens with 150 mm side lengths of each group were prepared, and the cube compressive strength test results are shown in Table 3.

The center pullout specimens were designed using cubic specimens with 150 mm side lengths according to ACI440.3R-12.(2012) [36]. The bond length were set to be 5d (60 mm) to avoid uneven distribution of bond stresses [37]. The unbonded area was separated by 90 mm smooth rigid PVC tubes, which were attached to the BFRP bar with adhesive tape and sealed with resin for 1 mm at both ends of the PVC tube. This was to precisely control the bond length and prevent local extrusion damage to the concrete at the loading end. A length of 20 mm was reserved at the free end to measure the slip. Steel tubes were used to anchor the loading end to avoid shear failure caused by the testing machine grip. The pull-out specimen was shown in Fig 4.

## Test setup

The specimens that required freeze-thaw were pre-soaked in this test, according to GB/T50082-2009 [38]. Immersed them in water which was (20±2˚C for 4d to reach the water absorption saturation state before the freeze-thaw cycle test started. The slow freezing method was used to evaluate the freeze-thaw performance of the specimens in this test. The temperature in the freeze-thaw test chamber was kept at (-20 ~ -18˚C), and the freezing time was 4 ~ 5h. After the specimen finished freezing, water at (18 ~ 20˚C) was added immediately and the specimen was submerged over at least 20 mm, turning it into a thawing condition and the thawing time was 4~5h. The control system ensured that the conversion time for freezing and thawing did not exceed 30 min. The end of 1 freeze-thaw cycle occurred when the specimen thawing was completed, and then the next cycle could start. The slow freeze-thaw equipment was shown in Fig 5.

The center pull-out test used an electro-hydraulic servo testing machine of model WDW-300KN, as shown in Fig 6. The center pullout specimen was placed in a reaction force test device, as shown in Fig 7. The loading end of the BFRP bar passed through the center hole of

**Table 3. Cube compressive strength test results.**

| Specimen number | Cube compressive strength (MPa) |
|---|---|
| FT0-C0B0 | 42.78 |
| FT25-C0B0 | 41.5 |
| FT50-C0B0 | 39.48 |
| FT75-C0B0 | 37.89 |
| FT100-C0B0 | 36.45 |
| FT0-C0.15B0.2 | 45.03 |
| FT25-C0.15B0.2 | 43.78 |
| FT50-C0.15B0.2 | 43.32 |
| FT75-C0.15B0.2 | 42.6 |
| FT100-C0.15B0.2 | 41.3 |

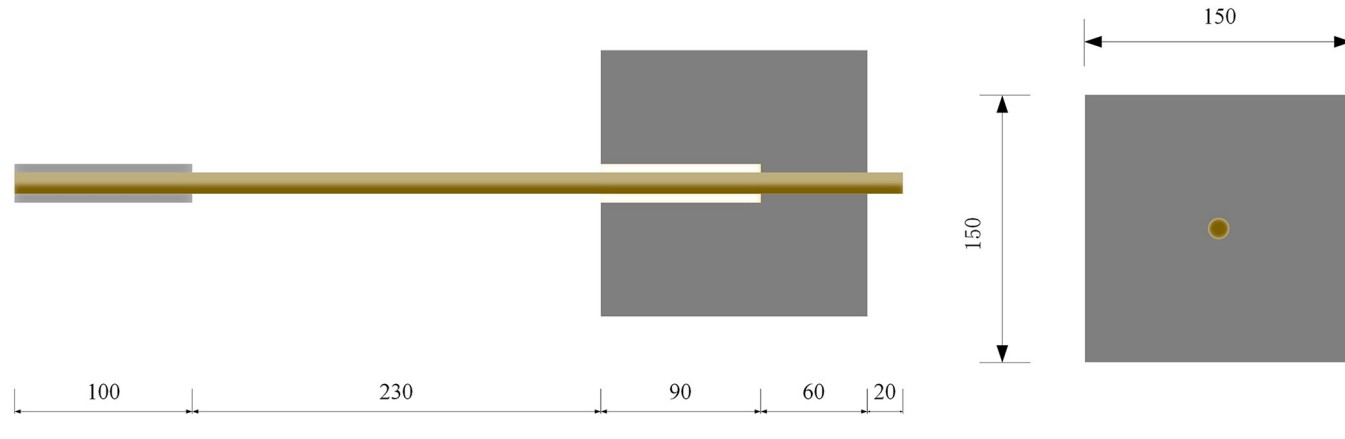

## (a) side view

## (b) front view

**Fig 4. Schematic diagram of specimen.**

the lower steel plate, and there was no contact between the bar and the steel plates. The center pullout test was controlled by displacement and the loading rate was set to 1mm/min.

The bond strength $\tau$ for each group of specimens was calculated according to ASTM D7913/D7913M-14 [39]. The formula as shown in Eq (1):

$$\tau = \frac{P}{\pi d l_a} \tag{1}$$

Where $\tau$ was the average bond stress between the BFRP bar and concrete, MPa; $P$ was the ultimate failure load of the specimen, N; $d$ was the diameter of the BFRP bar, mm; $l_a$ was the bond length of the BFRP bar, mm.

## Experimental results and discussion

### Test results and failure modes

Table 4 shows the pull-out test results. The failure modes were pull-out failure and splitting failure. All hybrid fiber-reinforced concrete specimens failed by pull-out, while ordinary concrete specimens failed by pull-out only with fewer freeze-thaw numbers.

**(1) Pull-out failure.** The specimens in this test mainly failed by pull-out. There was no crack on the concrete surface and no fracture on the bar, as shown in Fig 8(A). There was no slip at the beginning of the test. As the loading force increased, a small slip was generated on the loading end, while the free end slip remained unchanged. The bond stress was mainly provided by chemical adhesion at this stage. The loading end slip increased with the loading force, and the free end slip also started to increase. The composition of the bond stress changed to mechanical interaction and friction force as the chemical adhesion gradually disappeared. The slip increased rapidly and had a similar rate at both ends as the test continued. The loading force decreased and the slip rate accelerated until the loading force remained constant after it reached maximum value. Finally, the BFRP bars were pulled out. The failure process was stable and there was residual bond stress on the specimen after failure. The specimen was cut to observe the internal condition of the concrete after the test, as shown in Fig 8(B). The surface of the BFRP bars was severely damaged and had little concrete debris, and there were clear rib

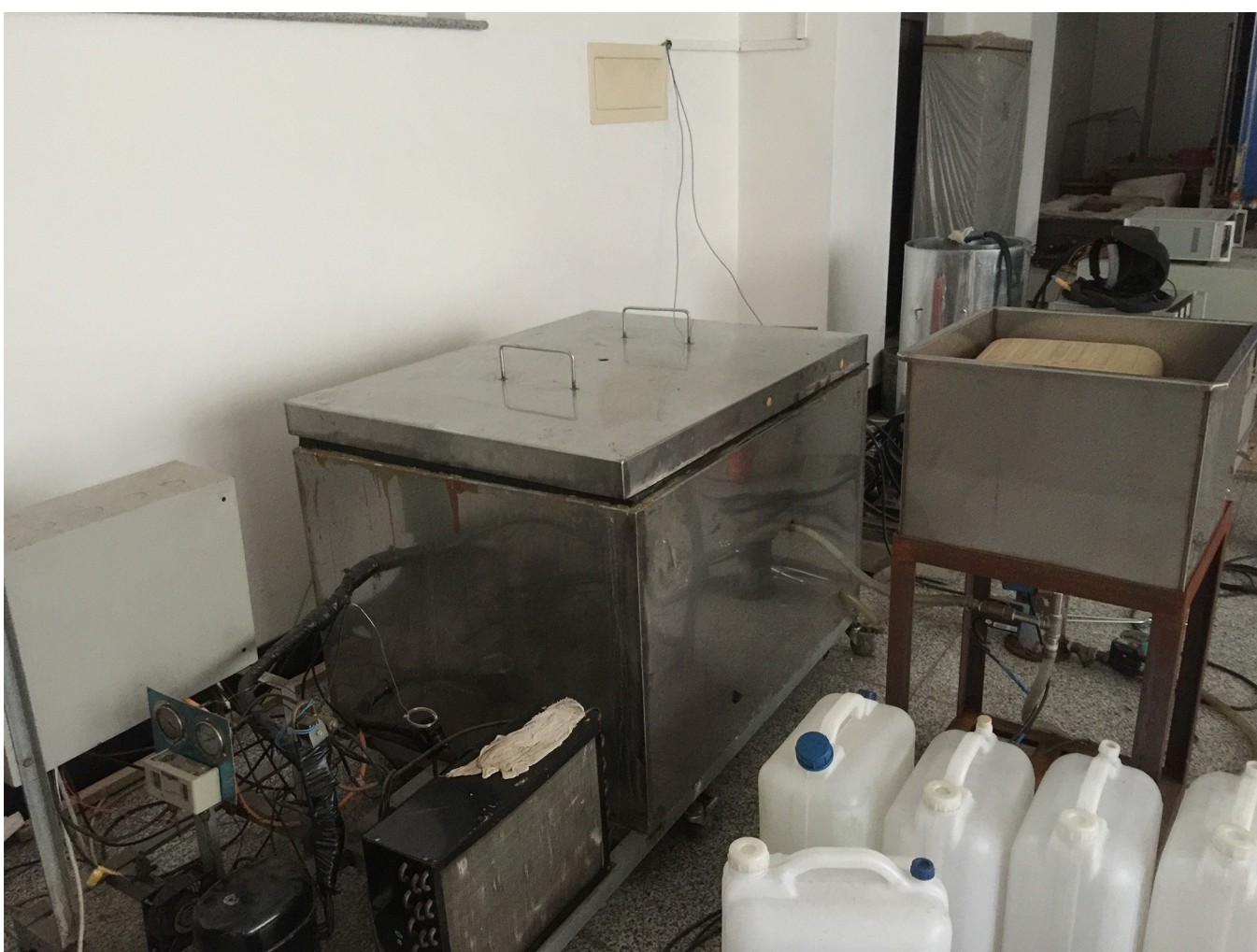

**Fig 5. Slow freeze-thaw equipment.**

marks at the concrete interface, indicating that the loading force damaged the chemical adhesion. Therefore, the mechanical interaction and the friction force had to resist the pull-out load together.

**(2) Splitting failure.** Splitting failure was radial cracking that was centered on the BFRP bar and the concrete was split into two or three pieces with a bursting sound, as shown in Fig 9. The bond strength was not fully developed when the specimen failed and the loading force dropped to 0 abruptly. This kind of failure mode had a characteristic of brittle failure and should be avoided in practical projects. The splitting failure resulted from the wedge effect produced by the oblique extrusion of the transverse ribs of the bar, which exerted oblique forces on the concrete. These forces had axial and radial components that induced compression and tension on the concrete, respectively. As the loading force increased, the transverse ribs of the BFRP bar were damaged and the circumferential tensile stress on the concrete surrounding the bar rose. When this stress surpassed the ultimate tensile strength of the concrete, cracks formed along the direction of the bar and propagated to the surface of the specimen, leading to specimen failure and loss of adhesion. The BFRP bar ordinary concrete specimens with more freeze-thaw cycles were more prone to splitting failure. The number of freeze-thaw cycles

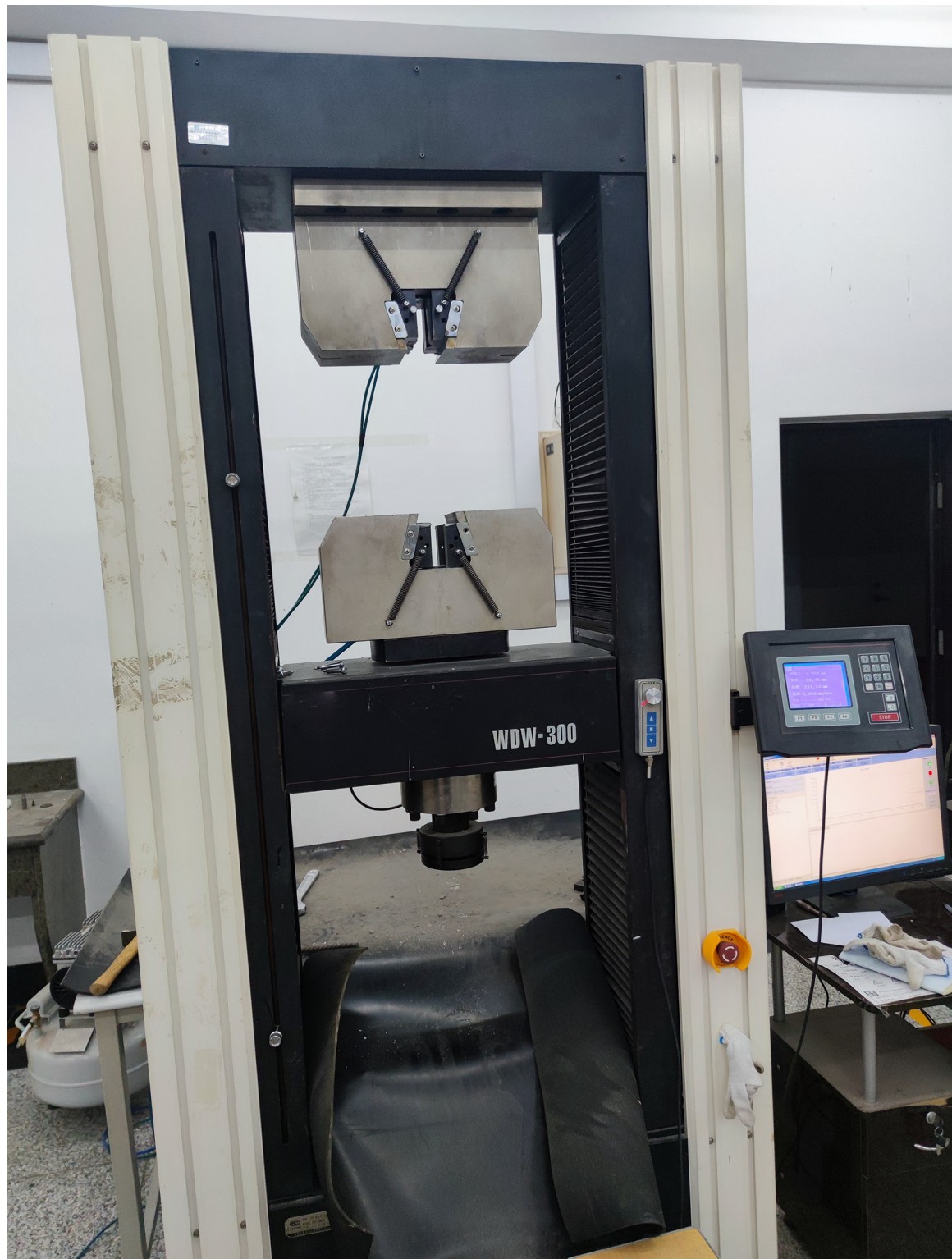

**Fig 6. Pull-out test machine.**

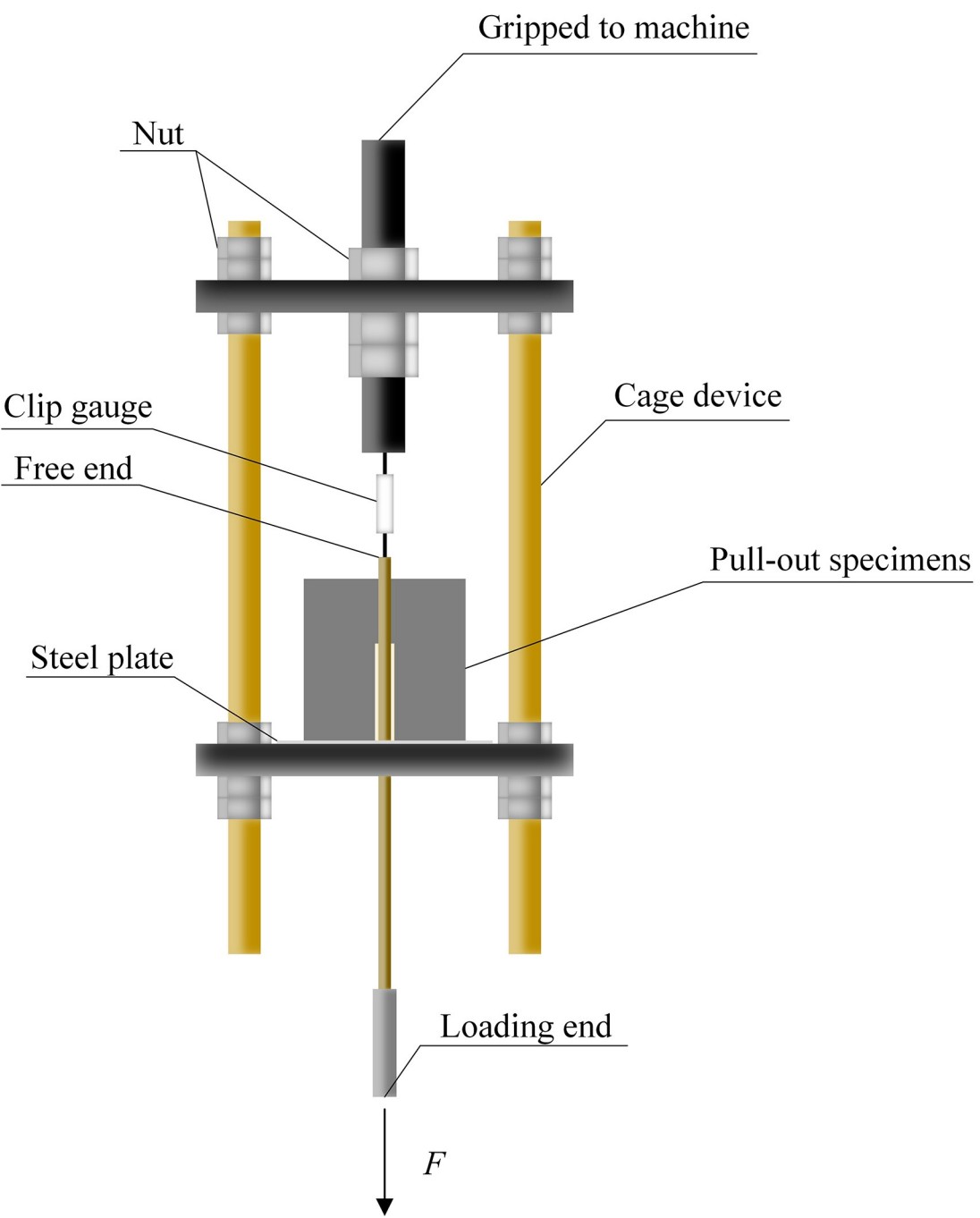

**Fig 7. Reaction force test device.**

affected the surface integrity of the specimens, causing the concrete to peel off and reducing the thickness of the concrete cover. This made the concrete less able to resist the stress component when the cover cracked [40]. The penetrating crack in the concrete was mainly caused by the "tension-compression" biaxial stress state, which resulted from the combination of the circumferential tensile stress from the transverse ribs on the BFRP bar and the compressive stress from the lower steel plates of the reaction force test device.

**Table 4. Pull-out test results.**

| Specimen number | $\tau_u$ | $S_u$ | Failure mode |
|---|---|---|---|
| | (MPa) | (mm) | |
| FT0-C0B0 | 21.57 | 2.29 | P |
| FT25-C0B0 | 19.62 | 3.3 | P |
| FT50-C0B0 | - | - | S |
| FT75-C0B0 | - | - | S |
| FT100-C0B0 | - | - | S |
| FT0-C0.15B0.2 | 23.27 | 2.87 | P |
| FT25-C0.15B0.2 | 20.45 | 3.55 | P |
| FT50-C0.15B0.2 | 18.82 | 3.78 | P |
| FT75-C0.15B0.2 | 16.18 | 4.04 | P |
| FT100-C0.15B0.2 | 12.71 | 5.03 | P |

$\tau_u$: Bond strength; $S_u$: Maximum slip under the peak load; P: Pull-out failure; S: Splitting failure.

## Bond-slip curve

Fig 10(A) and 10(B) show the bond-slip curves. The bond-slip curves of FT0-C0B0, FT25-C0B0, and all hybrid fiber-reinforced concrete specimens had four stages: micro slip, slip, pull-out, and residual stress. However, the bond-slip curves of FT50-C0B0, FT75-C0B0, and FT100-C0B0 only had the first two stages because these specimens failed by splitting and did not have descending sections on their curves. This was because of the freeze-thaw cycles that caused damage accumulation inside the concrete and concrete cover detachment. When the freeze-thaw action was more severe, cracks would be developed more during the pull-out process, weakening the mechanical interaction. Moreover, the reduction of the protective layer

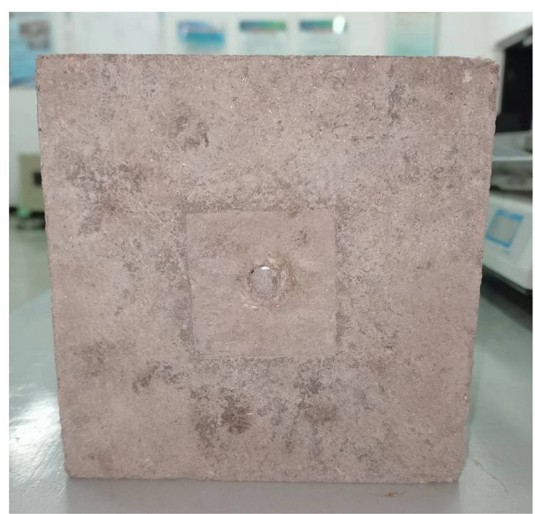

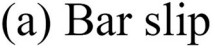
(a) Bar slip

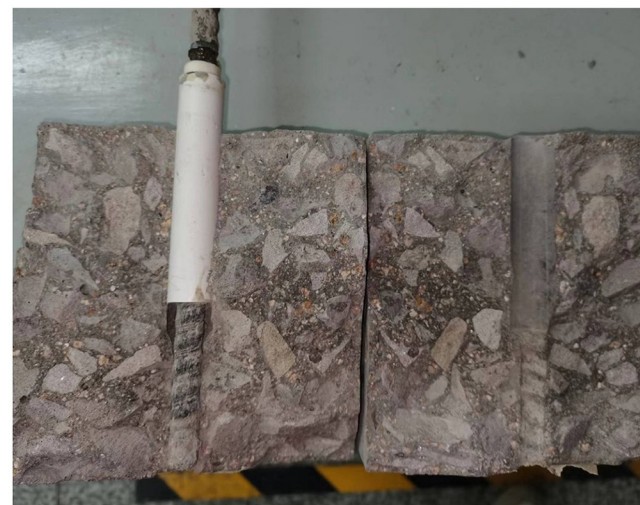
(b) Internal condition of the concrete and bar surface damage

**Fig 8. Pull-out failure.**

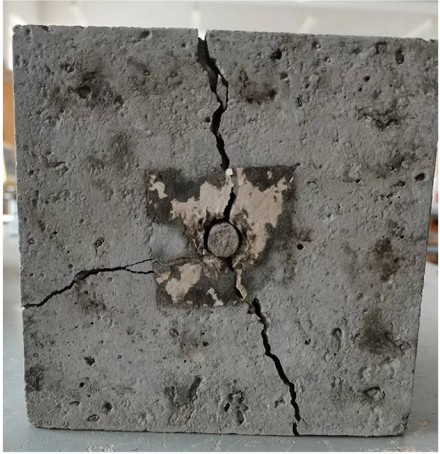 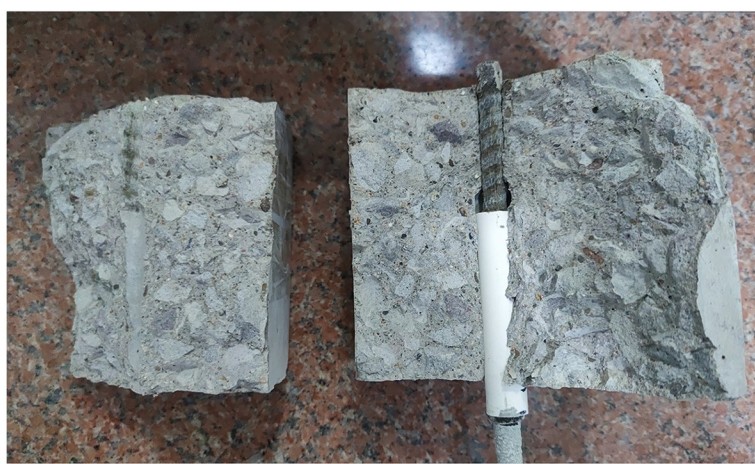

(a) Split of concrete    (b) Internal condition of the concrete and bar surface damage

**Fig 9. Splitting failure.**

impaired the concrete's ability to resist the stress components from the BFRP bars [41]. The specimen was destroyed before reaching the ultimate bond strength, resulting in the absence of a descending part of the curve due to these two adverse effects.

The first stage (micro slip stage): The curve was linear with a steep slope, indicating a rapid increase in bond stress with a small slip, and the bond stress was mainly derived from chemical adhesion at this time.

The second stage (slip stage): The curve reached the slip stage as the loading force increased, and a gradual decrease in bond stiffness and an increase in slip were caused. The bond stress attained its maximum value when the bond stiffness became 0. At this stage, the chemical adhesion diminished and the main composition of the bond stress was mechanical interaction and friction force. The specimens with 25 freeze-thaw cycles had lower bond strengths than

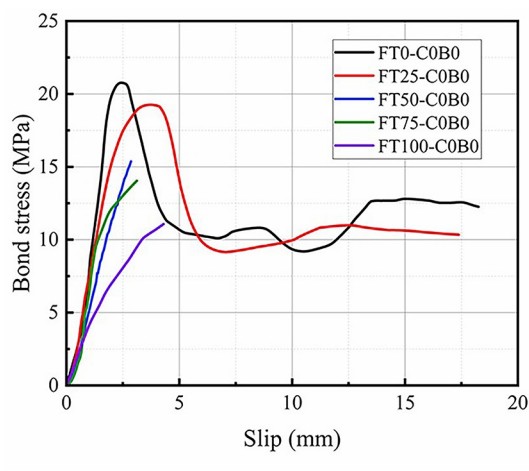 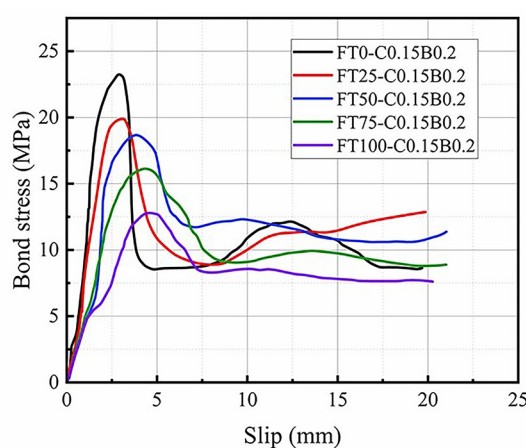

(a) Ordinary concrete    (b) Hybrid fiber-reinforced concrete

**Fig 10. Bond-slip curves.**

the specimens without freeze-thaw cycles. As we indicated, the damage inside the concrete of the specimens with 25 freeze-thaw cycles was greater and the peak load was significantly lower. Moreover, the slope of the curve at this stage showed a decreasing trend with the number of freeze-thaw cycles. This was because the freeze-thaw cycle continuously reduced the elastic modulus of specimens, and this process was not prevented by the addition of BF and CF. The bond stiffness of pull-out specimens slightly exceeded that of specimens subjected to 50 freeze-thaw cycles when the number of freeze-thaw cycles reached 75. This phenomenon could be attributed to the inclusion of certain inorganic solutes during the freeze-thaw test, which infiltrate the concrete [42]. As the freeze-thaw cycle count increased, unhydrated cement particles within the concrete react with these solutes and water, resulting in secondary hydration reactions and the generation of additional hydration products. Consequently, the internal structure of the concrete became denser, leading to a marginal increase in elastic modulus. Although this effect is subtle, the slight rebound in concrete's elastic modulus contributes to the slightly greater bond stiffness observed in pull-out specimens after 75 freeze-thaw cycles compared to those after 50 cycles.

The third stage (pull-out stage): The bond stress dropped rapidly when the bond stress reached its peak value, due to the wear of the ribs in contact with the concrete. At this point, the slip increased rapidly and the pull-out stage began. The bond stress at this stage was supplied by mechanical interaction and friction force, and the chemical adhesion vanished. The bond stress decreased more smoothly when the number of freeze-thaw cycles increased, indicating the better ductility of the specimen.

The fourth stage (residual stress stage): When the bond stress continued to decrease, the slip increased. The bond stress showed a periodic decay when the length of the bar pulled out reached a rib spacing, marking the start of the residual stress stage. A new mechanical anchorage force and a new bond stress maximum value emerged at this stage, but they were much lower than the maximum value of the second stage. This was because the BFRP bars and concrete were damaged by wear and tear during the pullout process and the damage accumulated.

## Bond strength

**Influence of freeze-thaw cycles on bond strength.** Groups of specimens of TF0-C0B0, TF25-C0B0, TF0-C0.15B0.2, TF25-C0.15B0.2, TF50-C0.15B0.2, TF75-C0.15B0.2, and TF100-C0.15B0 were used to study the effect of freeze-thaw cycles on the bond strength. Test results from 7 groups of specimens were compared as shown in Fig 11(A). The comparison revealed that the bond strength decreased with the increase in the number of freeze-thaw cycles. The specimens of BFRP bars with ordinary concrete had a 9.04% lower bond strength than the TF0-C0B0 specimens after 25 freeze-thaw cycles. The bond strength of hybrid fiber-reinforced concrete reduced by 12.11%, 19.12%, 30.47%, and 45.38%, respectively, after 25, 50, 75, and 100 freeze-thaw cycles compared with the TF0-C0.15B0.2 specimens. This was because the freeze-thaw cycle made the concrete loose and easy to peel off, increased the internal freeze-thaw damage, reduced the integrity of the concrete, and created cracks and pores in the concrete at the contact surface with BFRP bars. Moreover, the difference in the thermal expansion coefficients between bars and concrete [43] generated thermal stresses on the contact surfaces of concrete and BFRP bars. The BFRP bar had a greater shrinkage deformation than the concrete when the specimen was exposed to a low-temperature effect, and a gap at the interface between the bar and the concrete occurred. bond strength decreased as a result of the two negative effects superimposed on one another.

**Influence of hybrid fiber on bond strength.** Fig 11(B) shows a histogram of the effect of the incorporation of the hybrid fiber on bond strength. The bond strengths of hybrid fiber-

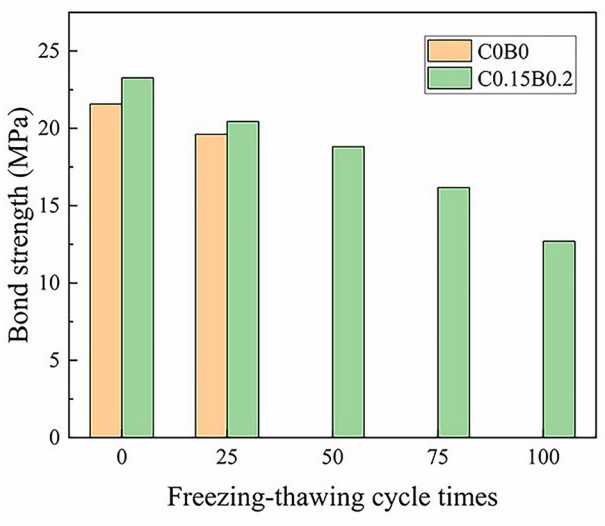

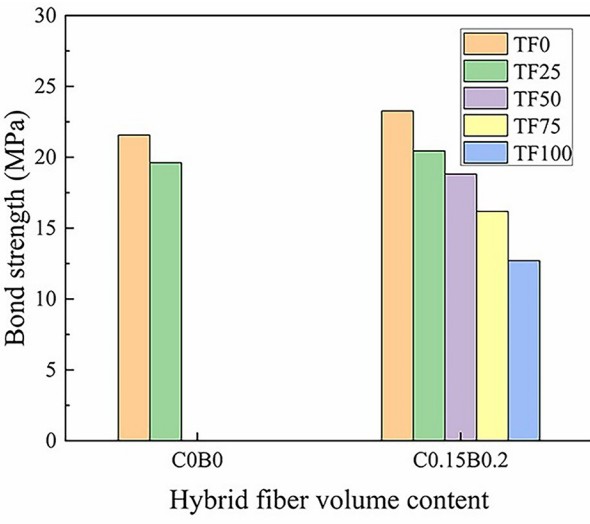

(a) Different freeze-thaw cycles

(b) Different fiber content

**Fig 11. Bond strength.**

reinforced concrete specimens were always higher than that of ordinary concrete specimens for the same number of freeze-thaw cycles. The bond strength of specimen C0.15B0.2 increased by 7.88% compared to that of specimen C0B0 under 0 freeze-thaw cycles. The bond strength of each specimen gradually decreased with an increasing number of freeze-thaw cycles. However, after 25 freeze-thaw cycles, the bond strength of specimen C0.15B0.2 remained 4.23% higher than that of specimen C0B0. Notably, the enhancement effect of hybrid fibers on bond strength diminished as the freeze-thaw cycle count increased. The effect of reinforcing toughness of the "fiber network" formed by the random distribution of CF and BF in concrete, which acted as a "micro-reinforcement". The initial defects of concrete could be compensated by the BF bridged at cracks within the concrete. The elastic modulus and tensile properties of BF surpassed those of cellulose fibers. Consequently, BF could more effectively enhance the tensile strength of concrete and restrain the propagation of internal cracks. Simultaneously, CF possessed a unique porous structure and natural hydrophilicity, allowing it to absorb non-frozen pore solution and free water within the concrete, thereby reducing internal hydrostatic pressure [29]. The uniform dispersion of CF and BF within the concrete matrix formed a three-dimensional spatial system. This not only elevated the overall tensile strength of concrete but also reduced the connectivity of internal pores, segmenting large pores into smaller ones and minimizing water infiltration. Consequently, concrete's frost resistance was effectively improved, while bond performance was maximally preserved. However, as the number of freeze-thaw cycles increased, internal freeze-thaw damage and the count of interconnected cracks escalated, leading to greater water ingress into the concrete. The proportion of moisture absorbed by CF and the voids segmented by hybrid fibers diminishes. Additionally, the $Ca(OH)_2$ produced during cement hydration dissolved in water. Water containing Ca$(OH)_2$ was absorbed into the CF's cavities, resulting in mineralization and a decline in the mechanical and durability properties of the fibers [44]. With more freeze-thaw cycles, an increasing amount of $Ca(OH)_2$ infiltrated the cavities, exacerbating fiber degradation. Consequently, the reinforcing effect of hybrid fibers gradually diminished. This phenomenon explained why, with an increasing number of freeze-thaw cycles, the reduction in bond

strength between hybrid fiber-reinforced concrete and BFRP bars exceeded that observed in ordinary concrete-BFRP bond strength.

## Bond-slip constitutive model between BFRP bar and concrete under the freeze-thaw cycle

**Existing bond-slip constitutive model of FRP bars.** The bond-slip constitutive model was an earlier method to predict the changing process of bond stress and slip between steel bars and concrete, and it had been widely recognized and adopted by scholars. With the development of FRP materials, FRP bars became more common in the construction industry, which made it crucial to predict the bond-slip process between FRP bars and concrete. Several bond-slip constitutive models of FRP bars with high recognition were developed as a result of continuous research by scholars [45]. Comparing the fitted bond-slip curve results from various models allowed us to assess the applicability of each bond-slip constitutive model in this experimental study. Subsequently, the most suitable bond-slip constitutive model for this experiment could be determined. This had significant engineering and research implications for predicting bond-slip curves.

The function expressions of each model were shown in Table 5. Malvar [46] proposed the first bond-slip constitutive model of GFRP bars after studying GFRP bars with three different surface treatments based on pull-out tests. The axisymmetrical radial pressure $\sigma_r$ was introduced, and the functional expression of $\sigma_r$ was defined by the peak bond stress $\tau_1$ and the peak slip $s_1$, which resulted in the complete functional expression of the bond-slip constitutive model being further derived. Eligehausen et al. [47] proposed the BPE model, which was first used to describe the bond-slip curve of deformed steel bars. Cosenza et al. [48] attempted to apply it to FRP bars, and successfully developed a BPE model applicable to FRP bars. The model consisted of four sections: an ascending stage, a horizontal stage, a descending stage, and a residual stage, as shown in Fig 12. However, the horizontal stage was found to be inconsistent with the bond-slip curve of FRP bars in many practical conditions as the research

**Table 5. Bond-slip constitutive model.**

| Constitutive model | Function expressions |
|---|---|
| Malvar model | $\tau_1/f_t = A + B(1 - e^{C\sigma_r/f_t}); s_1 = D + E\sigma_r;$ |
| | $\tau/\tau_1 = \frac{F(s/s_1)+(G-1)(s/s_1)^2}{1+(F-2)(s/s_1)+(G-1)(s/s_1)}$ |
| BPE model | $\tau/\tau_1 = (s/s_1)^\alpha, (s \leq s_1)$ |
| | $\tau = \tau_1, (s_1 < s \leq s_2)$ |
| | $\tau = \tau_1 - \frac{\tau_1 - \tau_3}{s_2 - s_3}(s_2 - s), (s_2 < s \leq s_3)$ |
| | $\tau = \tau_3 (s > s_3)$ |
| MBPE model | $\tau/\tau_1 = (s/s_1)^\alpha, (s \leq s_1);$ |
| | $\tau = [1 - p(s/s_1 - 1)]\tau_1, (s_1 < s \leq s_3);$ |
| | $\tau = \tau_3, (s > s_3)$ |
| CMR model | $\tau = (1 - e^{-s/s_r})^\beta \tau_1$ |
| Continuous curve model | $\tau = \left(2\sqrt{s/s_1} - \frac{s}{s_1}\right)\tau_1, (0 < s \leq s_1);$ |
| | $\tau = \frac{\tau_1(s_3-s)^2(2s+s_3-s_1)}{(s_3-s_1)^3} + \frac{\tau_r(s-s_1)^2(3s_3-2s-s_1)}{(s_3-s_1)^3}, (s_1 < s \leq s_3);$ |
| | $\tau = \tau_3, (s > s_3)$ |

$\sigma_r$: Axisymmetrical radial pressure; $f_t$: Tensile strength of concrete; $\tau_1$, $s_1$: Bond strength and corresponding slip; $\tau_3$, $s_3$: Residual strength and corresponding slip; $A$, $B$, $C$, $D$, $E$, $F$, $G$, $\alpha$, $\beta$, $s_2$, $s_r$: Parameters determined by test results.

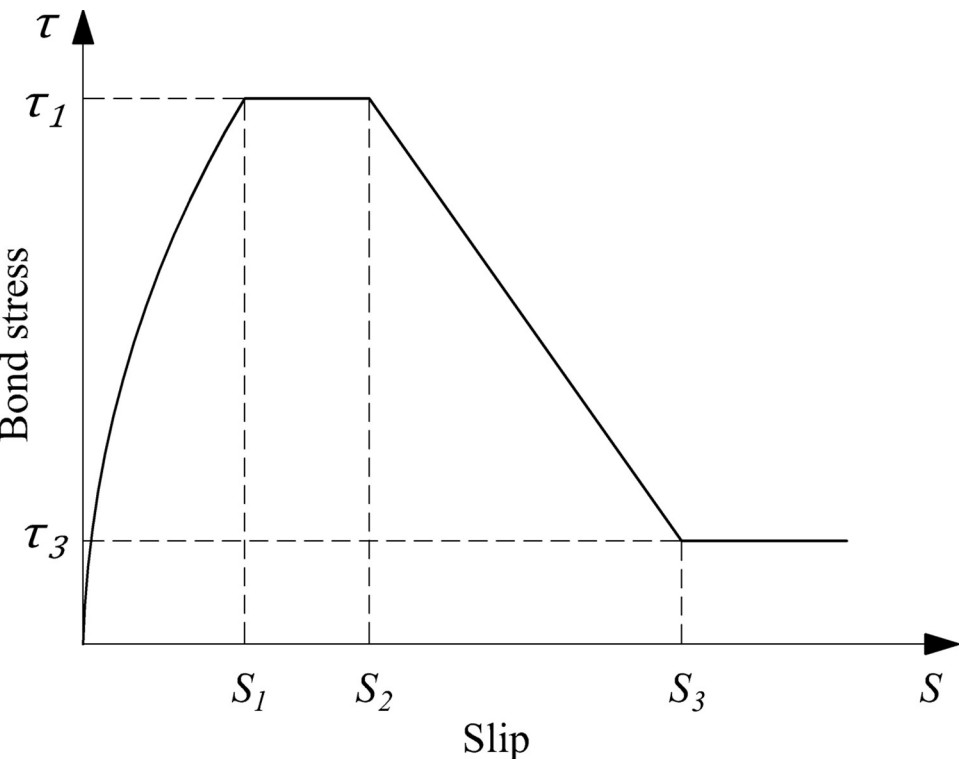

**Fig 12. BPE model.**

progressed. Therefore, Cosenza et al. [49] improved the BPE model, resulting in the MBPE model. The omission of the horizontal section and the modification of the functional expression of the descending section in the BPE model made the bond-slip constitutive curve more complete and more consistent with the practical conditions, as shown in Fig 13. It had been discovered that the accuracy of fitting the ascending section was required highly because the ascending section was the only section that must be taken into account in most structural calculations and practical applications. Thus, the CMR model which only had ascending section was proposed by Cosenza et al. [50]. The connection between each section of the above-mentioned bond-slip constitutive curves was not smooth enough and the inflection points were more noticeable, according to the research of Gao et al. [51]. As a result, a mathematical model of a continuous curve was proposed based on the above-mentioned models to make the curve more integral and smooth.

**Bond-slip constitutive models comparison.** The ascending sections of the bond-slip curves of BFRP bars obtained from this test were fitted by Origin software using four models: the Malvar model, MBPE model, CMR model, and Continuous Curve model. The average coefficient of determination (average $R^2$ values) were 0.991, 0.930, 0.979, and 0.816, respectively. Fig 14 shows a comparison of the $R^2$ values. It shows that the Malvar model and the CMR model could fit the ascending sections of the bond-slip curve accurately, with the Malvar model having a slightly stronger fitting effect than the CMR model. The fitting effect of the MBPE model was weaker than that of the Malvar model and CMR model, and the fitting effect of the Continuous Curve model was the weakest. Figs 15 and 16 show that each model had a bigger fluctuated fitting effect for the curve of ordinary concrete, and a stronger fitting effect for hybrid fiber-reinforced concrete than for ordinary concrete. This was because ordinary concrete was brittle, and there were many small changes in the curve, which made it difficult

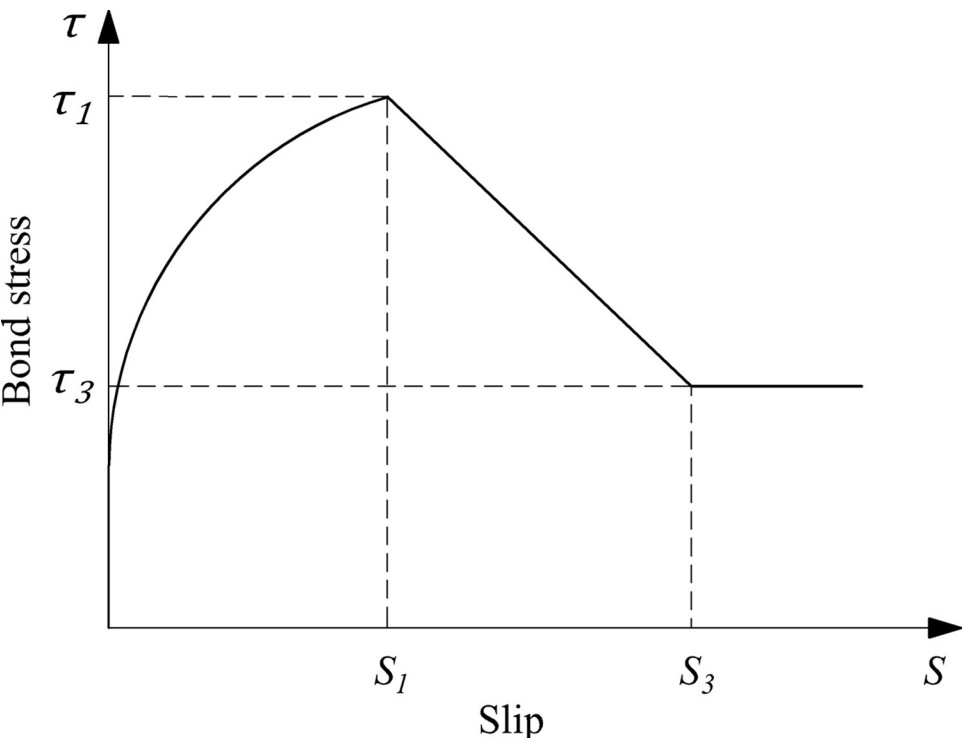

**Fig 13. MBPE model.**

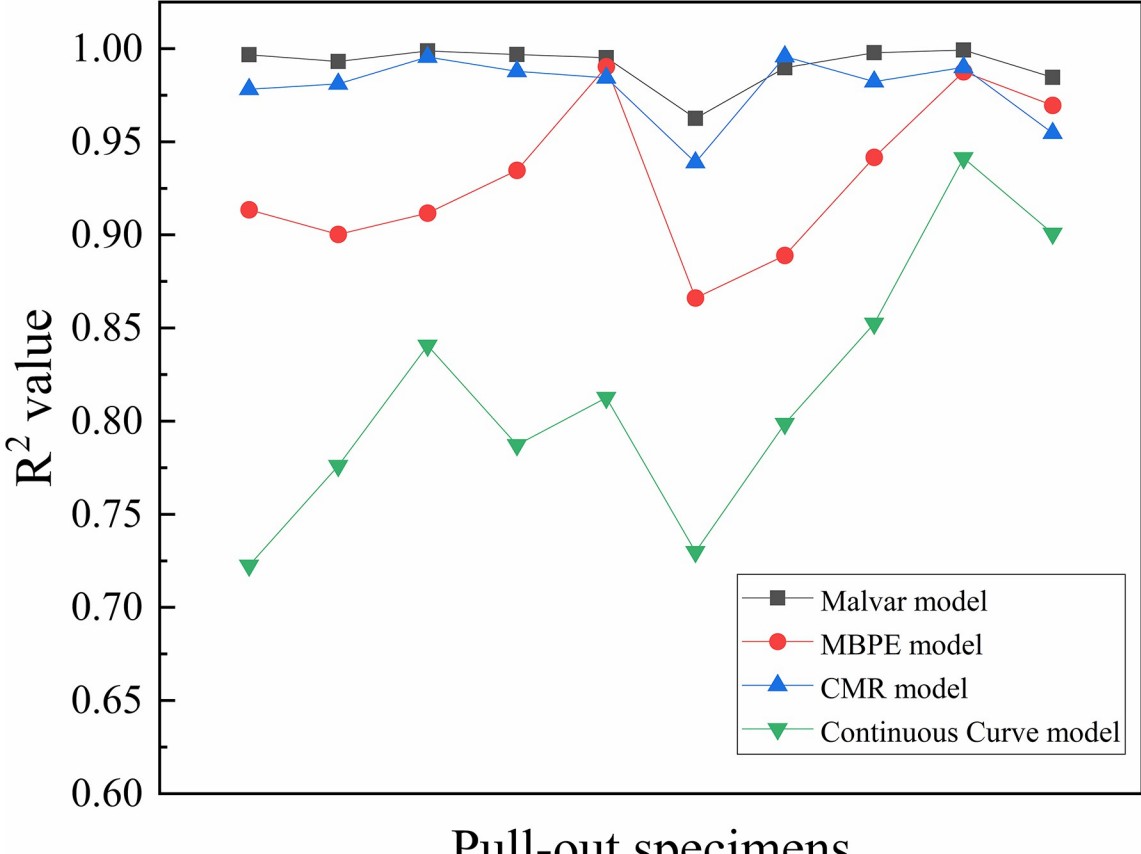

**Fig 14. Comparison of R$^2$ of ascending section.**

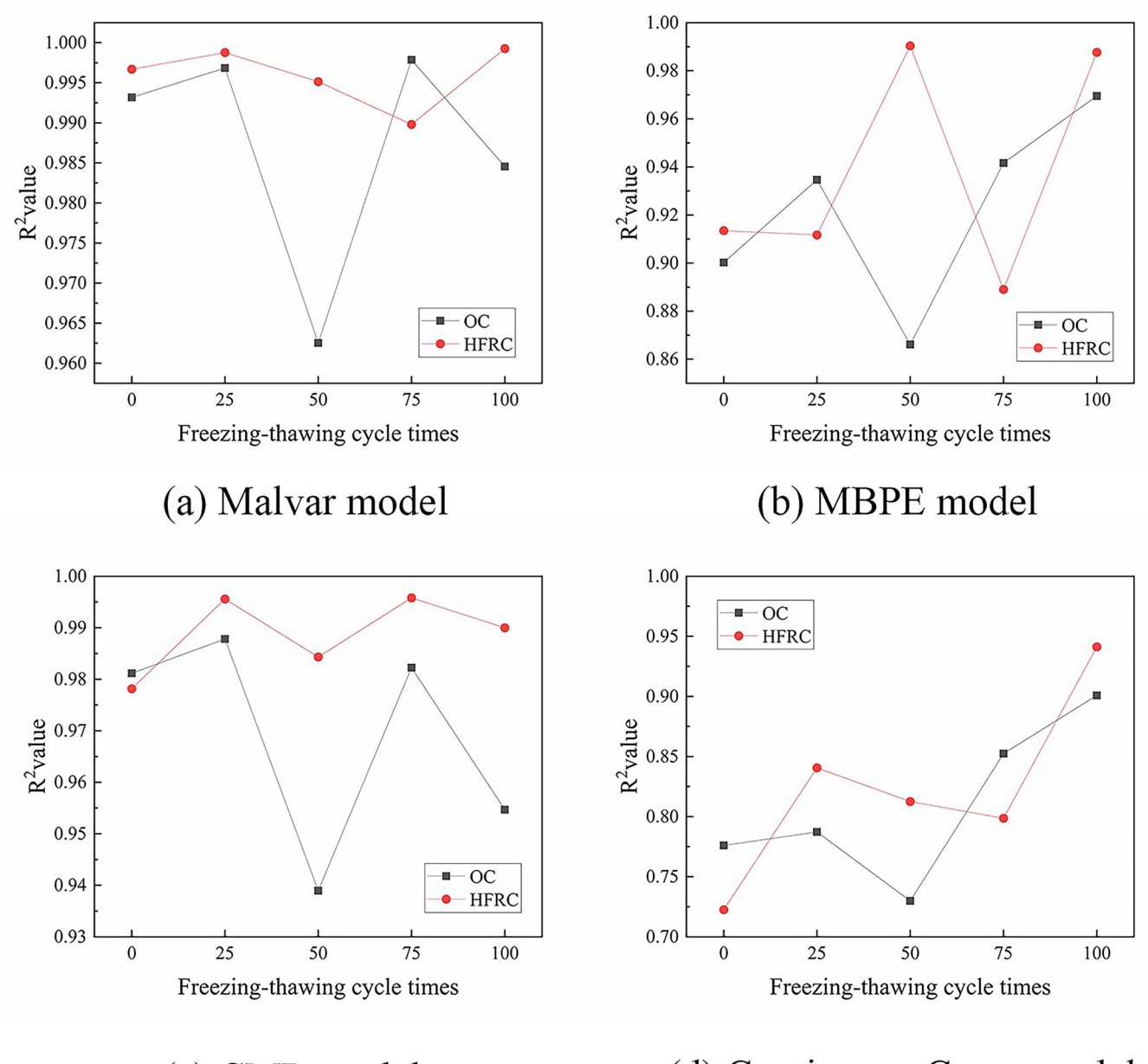

**Fig 15. R² values for each model fitting the ascending section.**

for each model to fit these small fluctuations accurately. It was also observed that the fitting effect of the MBPE model and the Continuous Curve model increased when the freeze-thaw cycles increased. This was because that the observed reduction in the slope of both the ascending and descending sections of the bond-slip curve, regardless of the inclusion of hybrid fibers, was attributed to the decrease in the elastic modulus of concrete induced by freeze-thaw cycles. Notably, bond-slip constitutive models often exhibit better fitting performance for curves with lower slopes.

The descending sections of the curves were fitted by three models: the Malvar model, MBPE model, and Continuous Curve model. The fitting effect of each model was analyzed

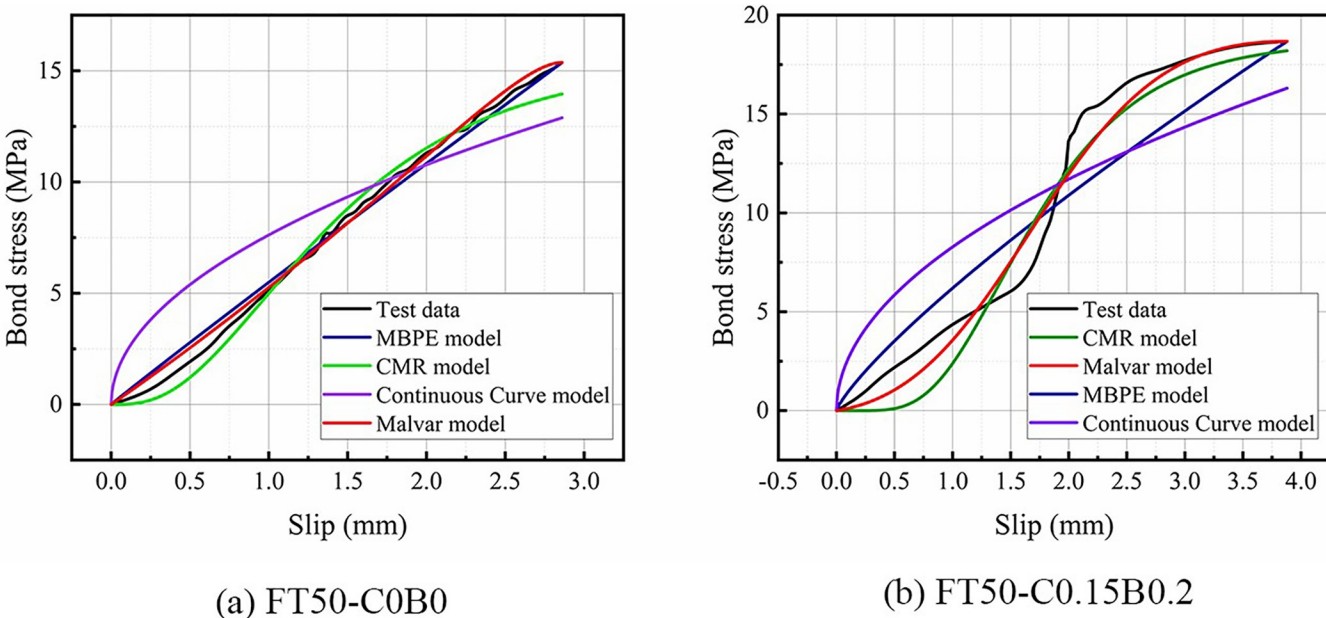

**Fig 16. Ascending section comparison.**

and compared, as shown in Fig 17. The average $R^2$ values of the Malvar model, MBPE model, and Continuous Curve model were 0.771, 0.930, and 0.971, respectively. The Continuous Curve model had an advantage in fitting the descending section compared to the ascending section, and it was better than both the Malvar model and the MBPE model. The MBPE model also had a good fitting effect, but slightly worse than the Continuous Curve model. However, the Malvar model had the worst fitting effect, which was completely different from its effect on the ascending sections. Figs 18 and 19 show that there was an overall increasing trend in the fitting effect of each model for hybrid fiber-reinforced concrete as the freeze-thaw cycles increased.

## Prediction of bond strength based on BP neural network

**BP neural network prediction model.** An information processing system called an artificial neural network (ANN) was built on an imitation of the brain's neural network structure and function. It had many advantages compared with the traditional semi-empirical method [52]. It could be divided into the feedforward network and the feedback network [53]. The backpropagation (BP) neural network was a forward neural network according to the error backpropagation algorithm, which was made up of the input layer, the hidden layer, and the output layer. There were several neurons in each of these layers, with the layers being fully interconnected (i.e., the units of the previous layer and the units of the next layer were connected to each other by a weight vector), and the connection between two neural units of the same layer being in the form of a break. It was a two-way information transfer. The computational model of neurons was shown in Eqs (2) and (3).

$$I = \sum\nolimits_{j=1}^{n} \omega_j x_i - \theta \tag{2}$$

$$y = f(I) \tag{3}$$

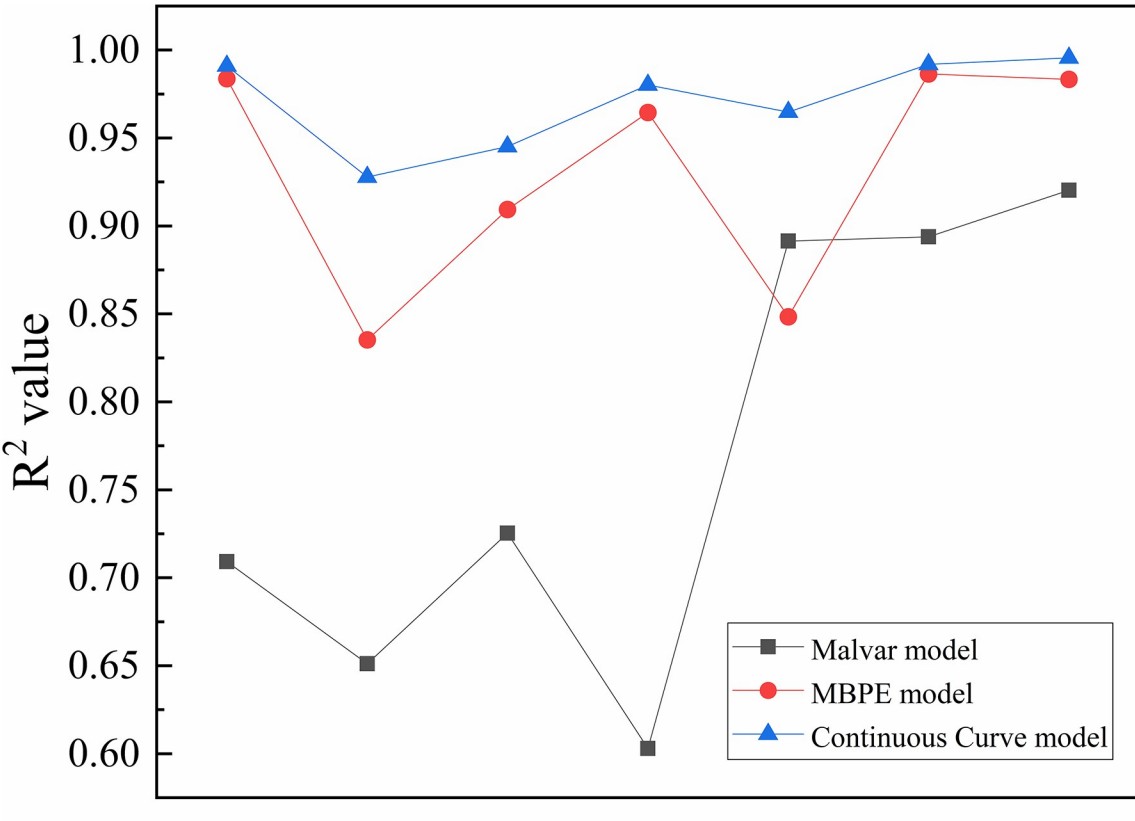

**Fig 17. Comparison of R$^2$ values of descending section.**

Where $x_i$ (= 1, 2, . . ., n) was the i th input signal in neurons; $\theta$ was the threshold value; $\omega_j$ was the weight coefficient representing the strength of the connection.

Establishing a mathematical model to precisely predict the bond strength was difficult because it was influenced by various factors and had a complex relationship with them. In addition, the test data was used by the traditional constitutive model to verify the fitting formula. In comparison to the traditional formula, a high nonlinearity was shown in the BP neural network prediction model in the prediction of bond strength, and part of the data was used for formula fitting while the rest of the data was used for verification, with higher prediction accuracy and more convincing results.

**Data processing and sample training.** A database of 96 groups of test results of FRP bars pull-out specimens [17, 33, 34, 54–57] was collected as training samples for a BP neural network in this investigation. A total of 10 factors, including the number of freeze-thaw cycles, types of FRP bars types, FRP bar diameter, surface treatment of bars, bond anchorage length, compressive strength of concrete, failure mode, BF volume content, CF volume content, and recycled aggregate replacement rate, were used as input parameters, with bond strength as an output parameter. The 10 input layer nodes, 13 hidden layer nodes, and 1 output layer node were set up respectively. The hyperbolic tangent S-type transfer function "tansig" was employed after the input data had been normalized using the function "mapminmax". The transfer function "tansig" was selected between the input and the hidden layers, and the transfer function "purelin" was selected between the hidden and output layers. The maximum

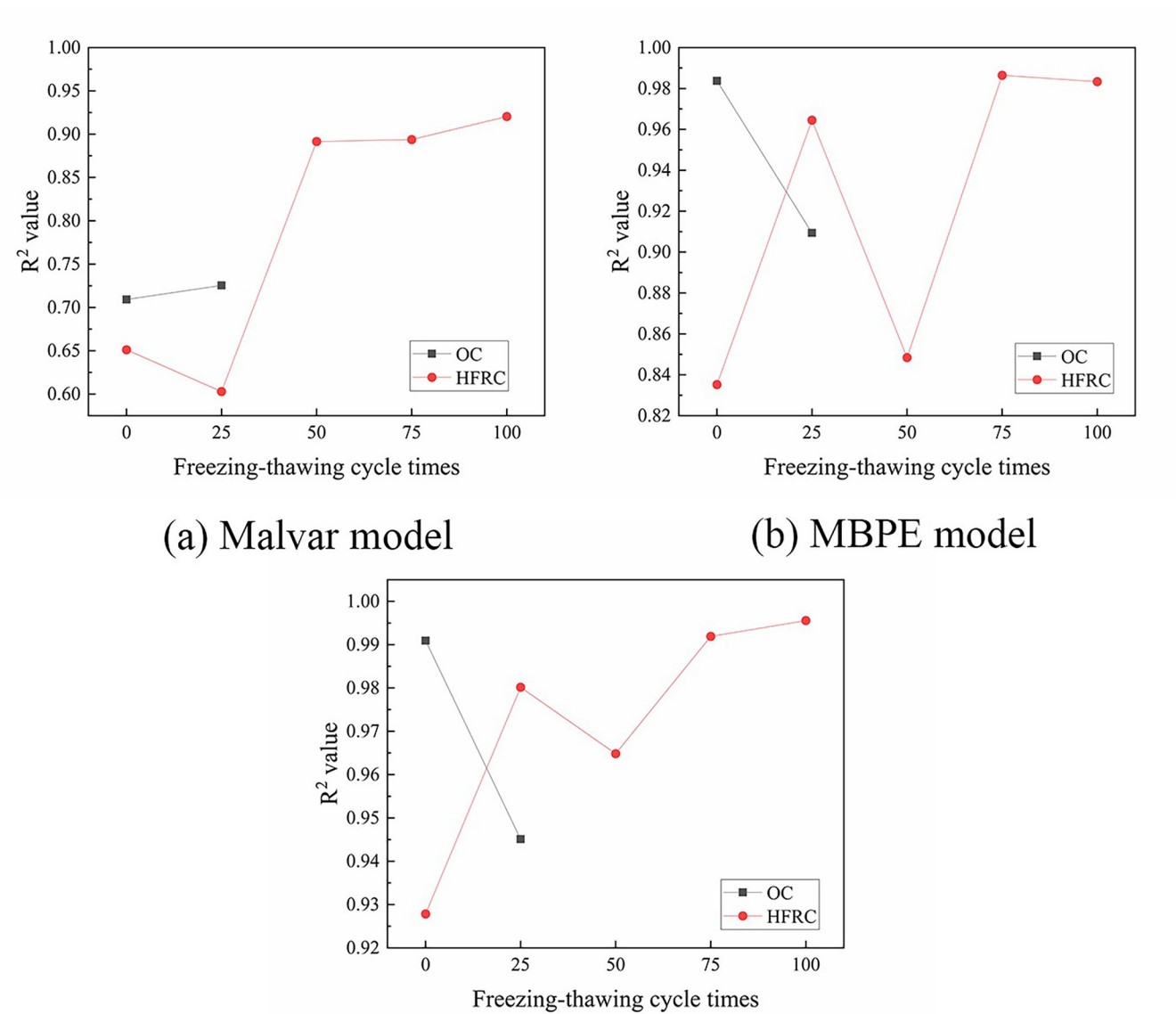

(a) Malvar model    (b) MBPE model

(c) Continuous Curve model

**Fig 18. R² values for each model fitting the descending section.**

number of training cycles (net.trainParam.epochs) was set to 3000, the number of display intervals (net.trainParam.show) was set to 25, the training goal (net.trainParam.goal) was set to $1 \times 10^{-5}$, the training time (net. trainParam.time) was set to Inf (unlimited time), the minimum performance gradient (net.trainParam.min_grad) was set to $1 \times 10^{-6}$, and the learning rate (net.trainParam.lr) was set to 0.01. Finally, the function "sim" was called to simulate and predict the trained model. BP neural network structure was shown in Fig 20.

The prediction system consisted of three parts: initialization, training, and simulation. The model samples were trained in the BP neural network structure which was established, and the feedback results are shown in Fig 21. The goodness-of-fit index R was close to 1 after the training of the sample data, which indicated that the training results were satisfactory, and the expected goal was achieved without over-fitting.

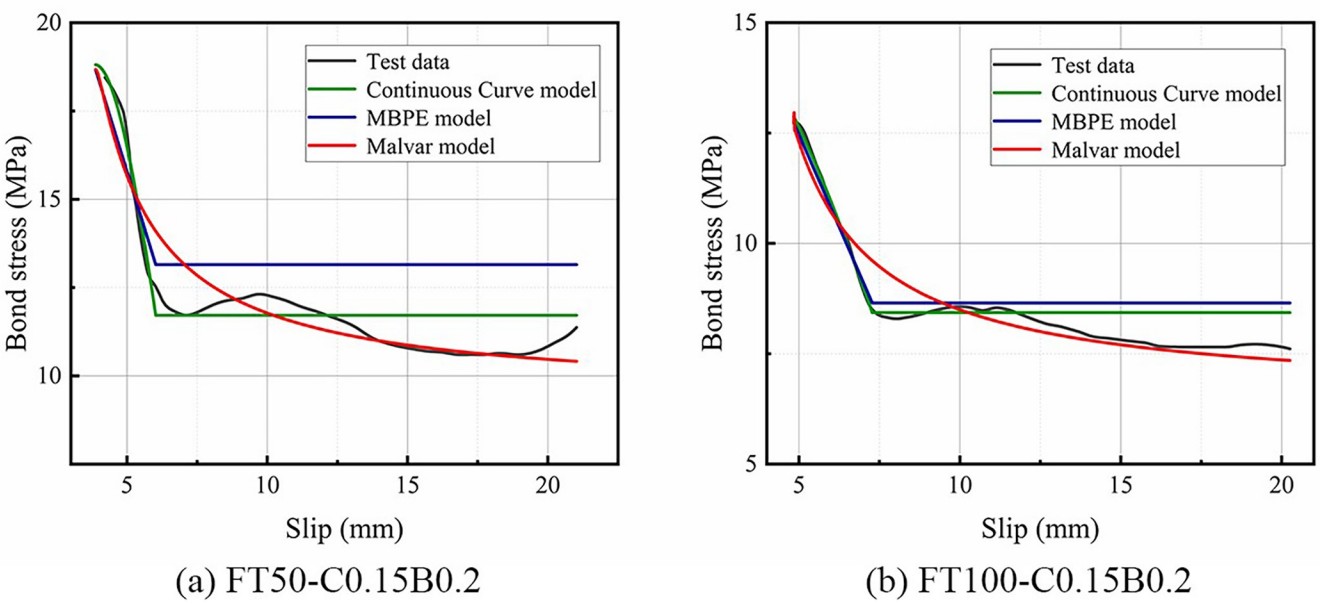

(a) FT50-C0.15B0.2

(b) FT100-C0.15B0.2

**Fig 19. Descending section comparison.**

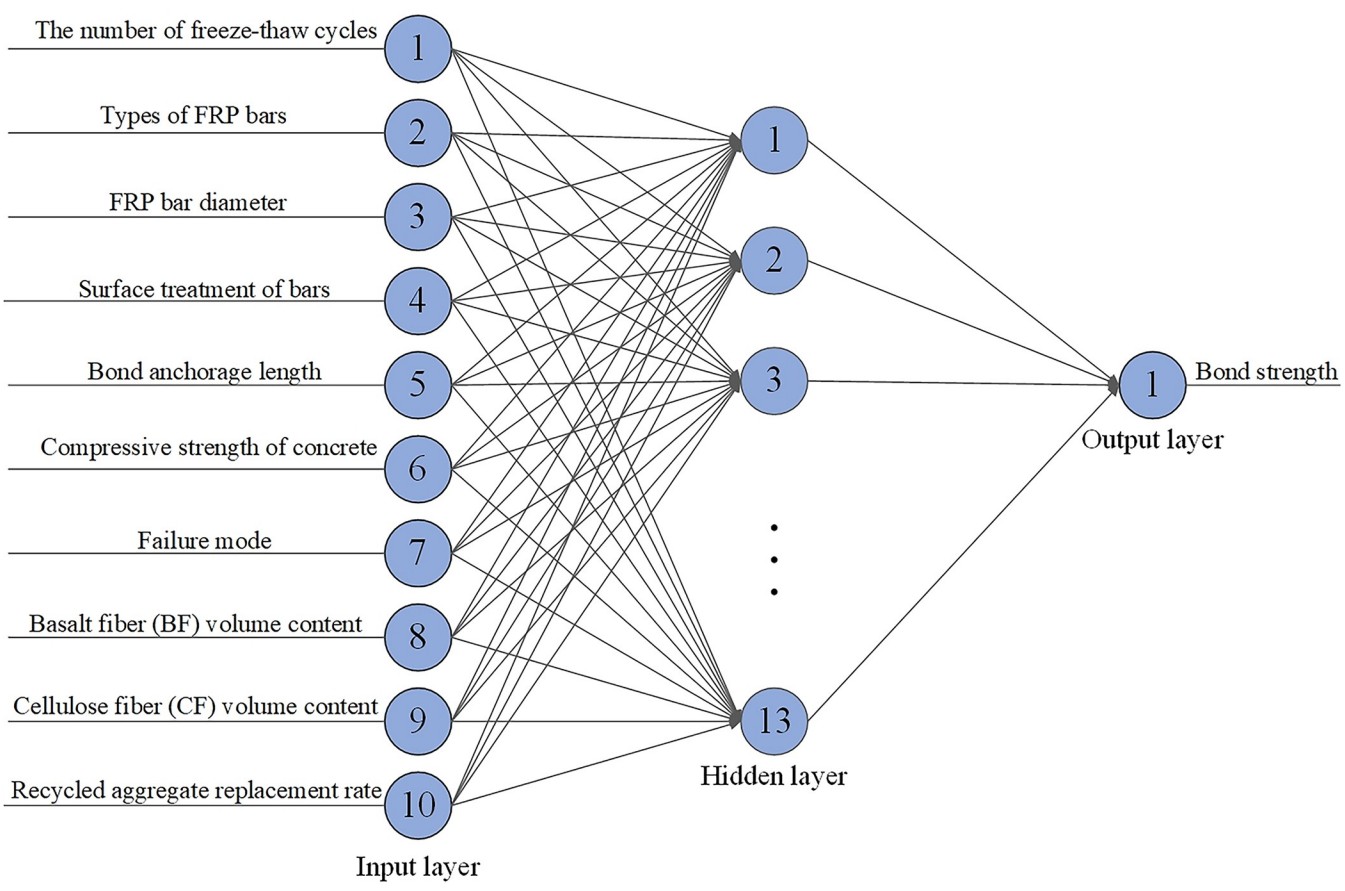

**Fig 20. BP neural network structure.**

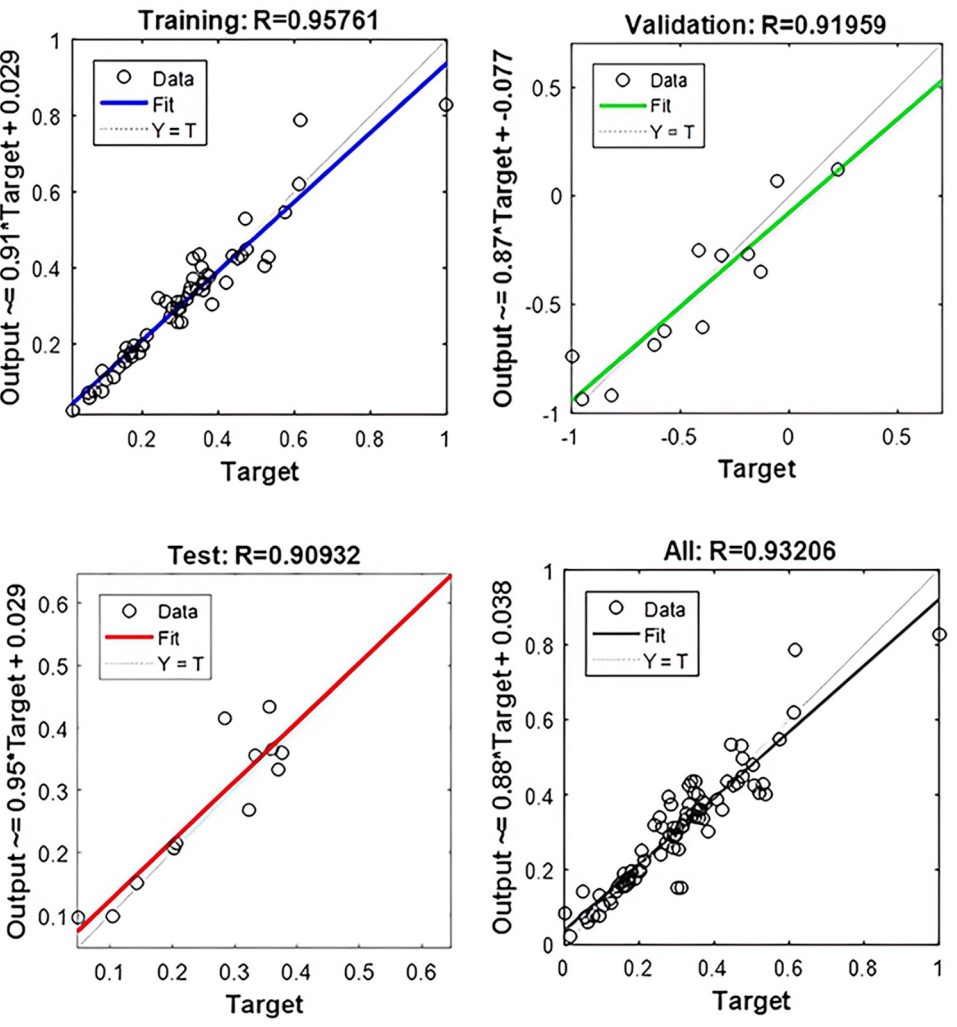

**Fig 21. Feedback of training results.**

**Analysis of model prediction results.** Fig 22 shows the comparison between the predicted data of the BP neural network and the test data, and Table 6 shows the predicted results obtained by training. The average $R^2$ value was 0.914, which indicated that the predicted results were satisfactory and the test data agreed well with the predicted data. The trained BP neural network model could predict the bond strength well, as evidenced by the relative errors, which ranged from 3.75% to 13.7%, of which 86% were less than 10%. The main reason for individual point values with errors exceeding 10% might be the lack of training samples near the point, which might cause insufficient training and inaccurate reflection of the mapping relationship between input and output by the sample space. Therefore, the BP neural network could accurately predict the bond strength and had strong promotion potential.

## Conclusions

This study investigated the bond performance between BFRP bars and hybrid fiber-reinforced concrete by experiment and theoretical analysis. It compared and analyzed the effects of the number of freeze-thaw cycles and the incorporation of hybrid fiber (BF0.2%, CF0.15%) on the

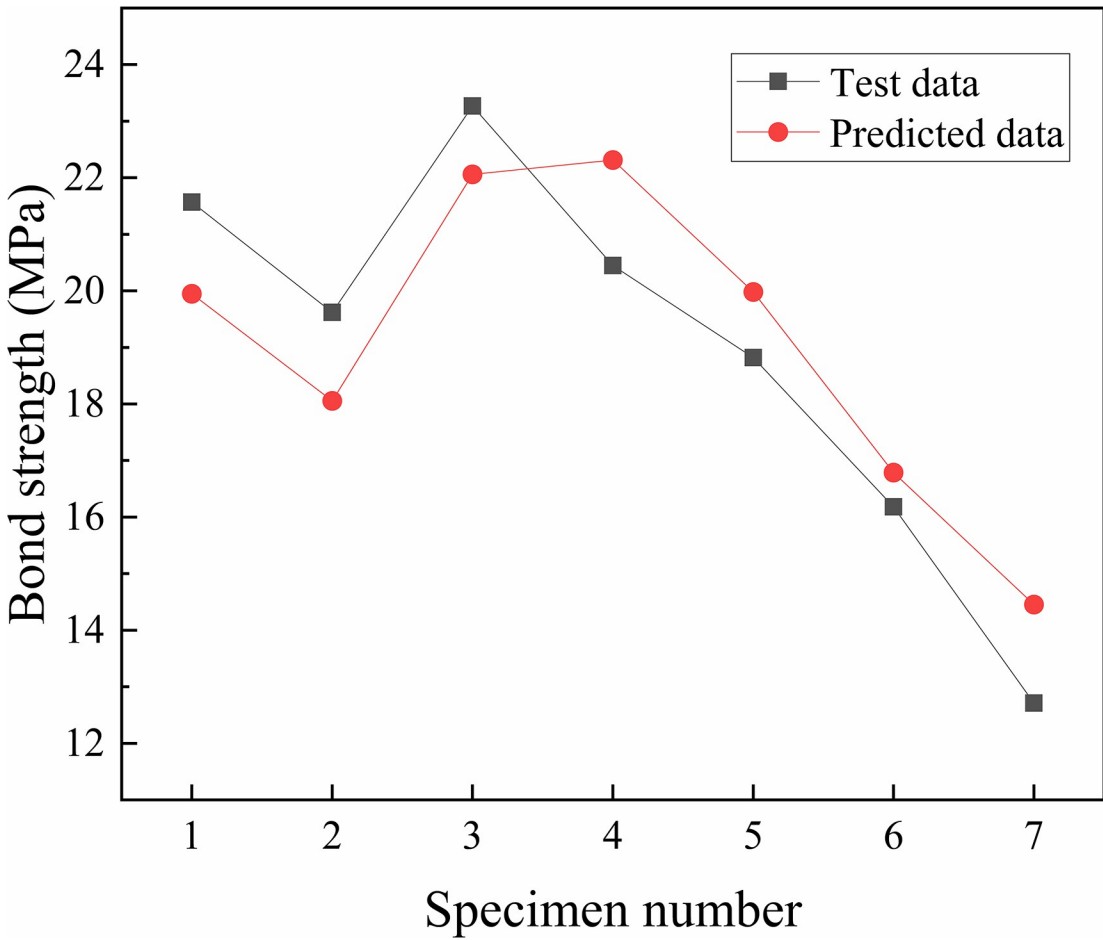

**Fig 22. Comparison between the predicted data of the BP neural network and the test data.**

bond strength, etc. The bond-slip constitutive model was used to fit the bond-slip curve and the results were compared, and a BP neural network was used to predict the bond strength. The following conclusions could be drawn.

1. The central pullout specimens after freeze-thaw cycles showed two failure modes. The first mode was pull-out failure, which produced complete bond-slip curves and better represented the bond characteristics between BFRP bars and concrete. The second mode was splitting failure, which resulted in incomplete bond-slip curves with only ascending

**Table 6. Comparison between test data and predicted results.**

| Number | Specimen number | Test data | Predicted data | Ratio of test data to predicted data | Error |
|---|---|---|---|---|---|
| 1 | FT0-C0B0 | 21.57 | 19.9493 | 1.08 | 7.5% |
| 2 | FT25-C0B0 | 19.62 | 18.0549 | 1.08 | 7.9% |
| 3 | FT0-C0.15B0.2 | 23.27 | 22.0598 | 1.05 | 5.2% |
| 4 | FT25-C0.15B0.2 | 20.45 | 22.3139 | 0.92 | 9.1% |
| 5 | FT50-C0.15B0.2 | 18.82 | 19.9822 | 0.96 | 6.1% |
| 6 | FT75-C0.15B0.2 | 16.18 | 16.7873 | 0.96 | 3.75% |
| 7 | FT100-C0.15B0.2 | 12.71 | 14.4557 | 0.88 | 13.7% |

sections. Moreover, the failure mode of ordinary concrete specimens changed from pull-out failure to splitting failure when the freeze-thaw cycles exceeded a certain number (n ≥ 50), while the failure mode of hybrid fiber-reinforced concrete specimens stayed as pull-out failure.

2. The bond strength of hybrid fiber-reinforced concrete specimens and ordinary concrete specimens both reduced as the number of freeze-thaw cycles increased. Meanwhile, the peak slip increased and the bond stiffness decreased, respectively.

3. The incorporation of BF and CF enhanced the bond strength of the hybrid fiber-reinforced concrete specimens compared to the specimens without fibers under the same freeze-thaw conditions. This was due to the hybrid fibers' ability to toughen and resist cracks as well as to reduce hydrostatic pressure and osmotic pressure.

4. The Malvar model fitted the ascending section of the bond-slip curve best, and each model fitted the ascending section of the curve of the hybrid fiber-reinforced concrete better than that of the ordinary concrete. The Continuous Curve model fitted the descending section best, and the fit of each model for the descending section of the curve of the hybrid fiber-reinforced concrete improved as the number of freeze-thaws increased. Compared to the bond-slip curve of pull-out specimens that have not undergone freeze-thaw cycles, employing the bond-slip constitutive model for predicting the bond-slip curve of specimens subjected to freeze-thaw cycles was expected to yield more accurate results.

5. BP neural network model sample data had a high goodness-of-fit index (R) close to 1 after the prediction. The trained model simulated the test data, and the relative error of the predicted bond strength ranged from 3.75% to 13.7%, with 86% of the error being less than 10%. This indicated that the simulation results were satisfactory.

## Supporting information

**S1 Data.**
(ZIP)

## Author Contributions

**Conceptualization:** Yanming Su.

**Data curation:** Yanming Su.

**Formal analysis:** Yanming Su.

**Funding acquisition:** Yanming Su.

**Investigation:** Yanming Su.

**Methodology:** Yanming Su.

**Project administration:** Yanming Su.

**Resources:** Yanming Su.

**Software:** Yanming Su.

**Supervision:** Yanming Su.

**Validation:** Yanming Su.

**Visualization:** Yanming Su.

**Writing – original draft:** Yanming Su.

**Writing – review & editing:** Yanming Su.

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
