## [Decision Letter · Decision Letter 0]

16 Jan 2024

PONE-D-23-40746Bond Performance between Hybrid Fiber-Reinforced Concrete and BFRP Bars under Freeze-thaw CyclePLOS ONE

Dear Dr. Su,

Thank you for submitting your manuscript to PLOS ONE. After careful consideration, we feel that it has merit but does not fully meet PLOS ONE’s publication criteria as it currently stands. Therefore, we invite you to submit a revised version of the manuscript that addresses the points raised during the review process.

**ACADEMIC EDITOR: **Endeavour to do a thorough revision of your article and  ensure that all the reviewers comments  are considered in detail

We look forward to receiving your revised manuscript.

Kind regards,

Paul Awoyera

Academic Editor

PLOS ONE

A clean copy of the edited manuscript (uploaded as the new *manuscript* file)”.

 [This work was supported by a grant from the National Natural Science Foundation of China (No. 51938009).].  

[This work was supported by a grant from the National Natural Science Foundation of China (No. 51938009).]

 [This work was supported by a grant from the National Natural Science Foundation of China (No. 51938009).].

6. We note that your Data Availability Statement is currently as follows: [All relevant data are within the manuscript and its Supporting Information files.]

7. PLOS requires an ORCID iD for the corresponding author in Editorial Manager on papers submitted after December 6th, 2016. Please ensure that you have an ORCID iD and that it is validated in Editorial Manager. To do this, go to ‘Update my Information’ (in the upper left-hand corner of the main menu), and click on the Fetch/Validate link next to the ORCID field. This will take you to the ORCID site and allow you to create a new iD or authenticate a pre-existing iD in Editorial Manager. Please see the following video for instructions on linking an ORCID iD to your Editorial Manager account: " ext-link-type="uri" xlink:type="simple">https://www.youtube.com/watch?v=_xcclfuvtxQ".

8. We note that Figure(s) 2, 3a, 3b, 5, 6, 7, 8a, 8b, 9a and 9b in your submission contain copyrighted images. All PLOS content is published under the Creative Commons Attribution License (CC BY 4.0), which means that the manuscript, images, and Supporting Information files will be freely available online, and any third party is permitted to access, download, copy, distribute, and use these materials in any way, even commercially, with proper attribution. For more information, see our copyright guidelines: http://journals.plos.org/plosone/s/licenses-and-copyright.

a. You may seek permission from the original copyright holder of Figure(s) 2, 3a, 3b, 5, 6, 7, 8a, 8b, 9a and 9b to publish the content specifically under the CC BY 4.0 license. 

Reviewers' comments:

Reviewer's Responses to Questions

**Comments to the Author**

1. Is the manuscript technically sound, and do the data support the conclusions?

Reviewer #1: Yes

Reviewer #2: Partly

2. Has the statistical analysis been performed appropriately and rigorously? 

Reviewer #1: No

Reviewer #2: Yes

3. Have the authors made all data underlying the findings in their manuscript fully available?

Reviewer #1: Yes

Reviewer #2: No

4. Is the manuscript presented in an intelligible fashion and written in standard English?

Reviewer #1: Yes

Reviewer #2: No

5. Review Comments to the Author

Reviewer #1: The topic discuss very important innovative material using BFRP into concrete as reinforcement while subjected to freeze and thawing. The topic has been discussed before by many entities; however, the new part might include the FRC there several issue might be considered to improve the article.

1) there are several references speaking about and interrupting with models that are not mentioned here

El-Nemr, A., Ahmed, E. A., Barris, C., Joyklad, P., Hussain, Q., Benmokrane, B. (2023). Bond performance of fiber reinforced polymer bars in normal- and high-strength concrete. In Construction and Building Materials (Vol. 393, p. 131957). Elsevier BV. https://doi.org/10.1016/j.conbuildmat.2023.131957

F. Faqih, T. Zayed, E. Soliman, Factors and Defects Analysis of Physical and Environmental Condition of Buildings, J. Build. Pathol. Rehabilit. 5 (2020) 1–15.

2) the figures are not in its position and need to be clearer

3) more analysis is required

Reviewer #2: The author investigated the impact of freeze-thaw cycles and hybrid fiber incorporation on the bond performance between BFRP (Basalt Fiber Reinforced Polymer) bars and concrete. The paper is generally comprehensive with clear language expression, and the research content holds significant engineering implications. However, there are still some controversial aspects that require the author to provide further clarification, modification, and supplementation:

1. In Section 2.1, regarding the description of the experimental materials, please provide additional details about the ribs of the BFRP, such as the rib spacing and rib height.

2. In Section 3.1, if the author intends to illustrate the impact of freeze-thaw cycles on the failure mode of specimens, please supplement corresponding failure diagrams for comparison.

3. In Section 3.2, based on the author's description, an increase in the number of freeze-thaw cycles is expected to result in a decrease in the bond-slip curve slope, i.e., the bond stiffness. Please explain the reason for the observed phenomenon where the bond stiffness of specimens subjected to 75 freeze-thaw cycles is greater than that of specimens subjected to 50 freeze-thaw cycles, both with and without fiber incorporation.

4. In Section 3.3, the correspondence between the two figures in Fig.11 and the analysis of the impact of freeze-thaw cycles and hybrid fibers on the bond strength between BFRP bars and concrete described in the article needs to be debated. Whether (a) it is more suitable to describe the effect of freeze-thaw cycles, and (b) it is more suitable to describe the effect of hybrid fiber addition.

5. The significance of hybrid fibers requires clarification, and comparing only to specimens without fibers lacks persuasiveness. If possible, a comparison with specimens containing only BF and CF would enhance the completeness of the experimental results.

6. The significance of comparing the fitting results of the bond slip model needs further explanation.

7. In Section 3.5.2, including the replacement rate of recycled aggregate as one of the influencing factors, seems unnecessary. Please clarify the relevance of choosing this parameter and its significance, as it appears less related to the content of this paper.

6. PLOS authors have the option to publish the peer review history of their article (what does this mean?). If published, this will include your full peer review and any attached files.

Reviewer #1: **Yes: **amr elnemr

Reviewer #2: No

---

## [Author Response · Author response to Decision Letter 0]

29 Mar 2024

Answer to Academic editor:

1. When submitting your revision, we need you to address these additional requirements. Please ensure that your manuscript meets PLOS ONE's style requirements, including those for file naming. The PLOS ONE style templates can be found at https://journals.plos.org/plosone/s/file?id=wjVg/PLOSOne_formatting_sample_main_body.pdf and https://journals.plos.org/plosone/s/file? id=ba62/PLOSOne_formatting_sample_title_authors_affiliations.pdf.

R: Thanks for your suggestion. I have revised the manuscript to meet PLOS ONE’s style requirements.

Upon resubmission, please provide the following: The name of the colleague or the details of the professional service that edited your manuscript A copy of your manuscript showing your changes by either highlighting them or using track changes (uploaded as a *supporting information* file) A clean copy of the edited manuscript (uploaded as the new *manuscript* file)”.

R: As the Reviewer's good instruction, I have tried our best to revise the English of the whole manuscript carefully. In order to make the whole manuscript better understanding, I have revised some long sentences into short sentences. Meanwhile, I also have asked some colleagues who are skilled of English language to help us for checking the English. I hope that the language is now acceptable for the next review process. Finally, if the revised article is still not up to the standard of your magazine, I would like to use the organisation you partner with to revise my article to meet your requirements Special thanks to you for your good comments !

3. Please note that PLOS ONE has specific guidelines on code sharing for submissions in which author-generated code underpins the findings in the manuscript. In these cases, all author-generated code must be made available without restrictions upon publication of the work. Please review our guidelines at https://journals.plos.org/plosone/s/materials-and-softwaresharing#loc-sharing-code and ensure that your code is shared in a way that follows best practice and facilitates reproducibility and reuse.

R: Thanks for your comment. The relevant data has been uploaded as Supporting Information files.

4. Thank you for stating the following financial disclosure: [This work was supported by a grant from the National Natural Science Foundation of China (No. 51938009).]. Please state what role the funders took in the study. If the funders had no role, please state: ""The funders had no role in study design, data collection and analysis, decision to publish, or preparation of the manuscript."" If this statement is not correct you must amend it as needed. Please include this amended Role of Funder statement in your cover letter; we will change the online submission form on your behalf.

R: Thanks for your suggestion. In the cover letter, I have explicitly stated that the funders had no role in study design, data collection and analysis, decision to publish, or preparation of the manuscript.

5. Thank you for stating the following in the Acknowledgments Section of your manuscript: [This work was supported by a grant from the National Natural Science Foundation of China (No. 51938009).] We note that you have provided funding information that is not currently declared in your Funding Statement. However, funding information should not appear in the Acknowledgments section or other areas of your manuscript. We will only publish funding information present in the Funding Statement section of the online submission form. Please remove any funding-related text from the manuscript and let us know how you would like to update your Funding Statement. Currently, your Funding Statement reads as follows: [This work was supported by a grant from the National Natural Science Foundation of China (No. 51938009).]. Please include your amended statements within your coverletter; we will change the online submission form on your behalf.

R: Thanks for your suggestion. I have already removed the information related to funding from the manuscript.

6. We note that your Data Availability Statement is currently as follows: [All relevant data are within the manuscript and its Supporting Information files.] Please confirm at this time whether or not your submission contains all raw data required to replicate the results of your study. Authors must share the “minimal data set” for their submission. PLOS defines the minimal data set to consist of the data required to replicate all study findings reported in the article, as well as related metadata and methods (https://journals.plos.org/plosone/s/data-availability#locminimal-data-set-definition). 

Authors do not need to submit their entire data set if only aportion of the data was used in the reported study.

If your submission does not contain these data, please either upload them as Supporting Information files or deposit them to a stable, public repository and provide us with the relevant URLs, DOIs, or accession numbers. For a list of recommended repositories, please see https://journals.plos.org/plosone/s/recommendedrepositories.

If there are ethical or legal restrictions on sharing a deidentified data set, please explain them in detail (e.g., data contain potentially sensitive information, data are owned by a third-party organization, etc.) and who has imposed them (e.g., an ethics committee). Please also provide contact information for a data access committee, ethics committee, or other institutional body to which data requests may be sent. If data are owned by a third party, please indicate howothers may request data access.

R: Thanks for your comment. The relevant data has been uploaded as Supporting Information files.

7. PLOS requires an ORCID iD for the corresponding author in Editorial Manager on papers submitted after December 6th, 2016. Please ensure that you have an ORCID iD and that it is validated in Editorial Manager. To do this, go to‘Update my Information’ (in the upper left-hand corner of the main menu), and click on the Fetch/Validate link next to the ORCID field. This will take you to the ORCID site and allow you to create a new iD or authenticate a pre-existing iD in Editorial Manager. Please see the following video for instructions on linking an ORCID iD to your Editorial Manager account: https://www.youtube.com/watch?v=_xcclfuvtxQ".

R: Thanks for your comment. I have updated and validated my ORCID in Editorial Manager.

8. We note that Figure(s) 2, 3a, 3b, 5, 6, 7, 8a, 8b, 9a and 9b in your submission contain copyrighted images. All PLOS content is published under the Creative Commons Attribution License (CC BY 4.0), which means that the manuscript, images, and Supporting Information files will be freely available online, and any third party is permitted to access, download, copy, distribute, and use these materials in anyway, even commercially, with proper attribution. For more information, see our copyright guidelines: http://journals.plos.org/plosone/s/licenses-and-copyright.

We require you to either (1) present written permission from the copyright holder to publish these figures specifically under the CC BY 4.0 license, or (2) remove the figures from your submission: a. You may seek permission from the original copyright holder of Figure(s) 2, 3a, 3b, 5, 6, 7, 8a, 8b, 9a and 9b to publish the content specifically under the CC BY 4.0 license. 

We recommend that you contact the original copyright holder with the Content Permission Form (http://journals.plos.org/plosone/s/file?id=7c09/contentpermission-form.pdf) and the following text:“I request permission for the open-access journal PLOS ONE to publish XXX under the Creative Commons Attribution License (CCAL) CC BY 4.0 (http://creativecommons.org/licenses/by/4.0/). 

Please be aware that this license allows unrestricted use and distribution, even commercially, by third parties. Please reply and provide explicit written permission to publish XXX under a CC BY license and complete the attached form.” Please upload the completed Content Permission Form or other proof of granted permissions as an ""Other"" file with your submission. 

R: Thank you for your advice. Figure (s) 2, 3a, 3b, 5, 6, 7, 8a, 8b, 9a and 9b all belong to the real photos I took during the experiment, and do not have copyright issues with other organizations or individuals. I still provide a Content Permission Form about myself.

Answer to Reviewer #1:

1. There are several references speaking about and interrupting with models that are not mentioned here El-Nemr, A., Ahmed, E. A., Barris, C., Joyklad, P., Hussain, Q., Benmokrane, B. (2023). Bond performance of fiber reinforced polymer bars in normal- and high-strength concrete. In Construction and Building Materials (Vol. 393, p. 131957). Elsevier BV. https://doi.org/10.1016/j.conbuildmat.2023.131957 F. Faqih, T. Zayed, E. Soliman, Factors and Defects Analysis of Physical and Environmental Condition of Buildings, J. Build. Pathol. Rehabilit. 5 (2020) 1–15.

R: Grateful for your valuable feedback, I acknowledge the meticulous scientific detail and substantial research significance embedded within the two referenced articles. My validation process involved scrutinizing the models delineated in these papers, leveraging data gleaned from my experimental study. Regrettably, the model’s applicability to the bond performance between BFRP reinforcement bars and hybrid fiber-reinforced concrete post freeze-thaw cycles proved inadequate. Nonetheless, I dutifully cite these seminal works to disseminate their pivotal findings.

[R1] El-Nemr A, Ahmed EA, Barris C, Joyklad P, Hussain Q, Benmokrane B. Bond performance of fiber reinforced polymer bars in normal- and high-strength concrete. Constr Build Mater. 2023;393:131957. doi: 10.1016/j.conbuildmat.2023.131957.

[R2] Faqih F, Zayed T, Soliman E. Factors and defects analysis of physical and environmental condition of buildings. J. Build. Pathol. Rehabilit. 2020;5(1):19. doi: 10.1007/s41024-020-00084-0.

(See Page 28, Lines 573-575; Pages 28, Lines 567-569)

2. The figures are not in its position and need to be clearer.

R: Appreciation is extended to the reviewer for their valuable suggestion. As per the submission guidelines outlined by the PLOS ONE journal, figures are expressly excluded from the main manuscript file. Instead, each figure must be meticulously prepared and submitted as an individual file. Notably, the image clarity has been further enhanced to align with the stringent requirements stipulated by the PLOS ONE journal.

3. More analysis is required.

R: I sincerely thank the reviewer for careful reading. I have incorporated various analyses, including the impact analysis of hybrid fibers on bond strength. As follows: “Fig 11(b) shows a histogram of the effect of the incorporation of the hybrid fiber on bond strength. The bond strengths of hybrid fiber-reinforced concrete specimens were always higher than that of ordinary concrete specimens for the same number of freeze-thaw cycles. The bond strength of specimen C0.15B0.2 increased by 7.88% compared to that of specimen C0B0 under 0 freeze-thaw cycles. The bond strength of each specimen gradually decreased with an increasing number of freeze-thaw cycles. However, after 25 freeze-thaw cycles, the bond strength of specimen C0.15B0.2 remained 4.23% higher than that of specimen C0B0. Notably, the enhancement effect of hybrid fibers on bond strength diminished as the freeze-thaw cycle count increased. The effect of reinforcing toughness of the “fiber network” formed by the random distribution of CF and BF in concrete, which acted as a “micro-reinforcement”. The initial defects of concrete could be compensated by the BF bridged at cracks within the concrete. The elastic modulus and tensile properties of BF surpassed those of cellulose fibers. Consequently, BF could more effectively enhance the tensile strength of concrete and restrain the propagation of internal cracks. Simultaneously, CF possessed a unique porous structure and natural hydrophilicity, allowing it to absorb non-frozen pore solution and free water within the concrete, thereby reducing internal hydrostatic pressure [R3]. The uniform dispersion of CF and BF within the concrete matrix formed a three-dimensional spatial system. This not only elevated the overall tensile strength of concrete but also reduced the connectivity of internal pores, segmenting large pores into smaller ones and minimizing water infiltration. Consequently, concrete’s frost resistance was effectively improved, while bond performance was maximally preserved. However, as the number of freeze-thaw cycles increased, internal freeze-thaw damage and the count of interconnected cracks escalated, leading to greater water ingress into the concrete. The proportion of moisture absorbed by CF and the voids segmented by hybrid fibers diminishes. Additionally, the Ca(OH)2 produced during cement hydration dissolved in water. Water containing Ca(OH)2 was absorbed into the CF’s cavities, resulting in mineralization and a decline in the mechanical and durability properties of the fibers [R4]. With more freeze-thaw cycles, an increasing amount of Ca(OH)2 infiltrated the cavities, exacerbating fiber degradation. Consequently, the reinforcing effect of hybrid fibers gradually diminished. This phenomenon explained why, with an increasing number of freeze-thaw cycles, the reduction in bond strength between hybrid fiber-reinforced concrete and BFRP bars exceeded that observed in ordinary concrete-BFRP bond strength. New analysis on the bond-slip curve, which was “The bond stiffness of pull-out specimens slightly exceeded that of specimens subjected to 50 freeze-thaw cycles when the number of freeze-thaw cycles reached 75. This phenomenon could be attributed to the inclusion of certain inorganic solutes during the freeze-thaw test, which infiltrate the concrete [R5]. As the freeze-thaw cycle count increased, unhydrated cement particles within the concrete react with these solutes and water, resulting in secondary hydration reactions and the generation of additional hydration products. Consequently, the i

---

## [Decision Letter · Decision Letter 1]

24 Apr 2024

Bond Performance between Hybrid Fiber-Reinforced Concrete and BFRP Bars under Freeze-thaw Cycle

PONE-D-23-40746R1

Dear Dr. Su,

We’re pleased to inform you that your manuscript has been judged scientifically suitable for publication and will be formally accepted for publication once it meets all outstanding technical requirements.

Kind regards,

Paul Awoyera

Academic Editor

PLOS ONE

Additional Editor Comments (optional):

Reviewers' comments:

Reviewer's Responses to Questions

**Comments to the Author**

1. If the authors have adequately addressed your comments raised in a previous round of review and you feel that this manuscript is now acceptable for publication, you may indicate that here to bypass the “Comments to the Author” section, enter your conflict of interest statement in the “Confidential to Editor” section, and submit your "Accept" recommendation.

Reviewer #1: All comments have been addressed

Reviewer #2: All comments have been addressed

2. Is the manuscript technically sound, and do the data support the conclusions?

Reviewer #1: Yes

Reviewer #2: Yes

3. Has the statistical analysis been performed appropriately and rigorously? 

Reviewer #1: N/A

Reviewer #2: Yes

4. Have the authors made all data underlying the findings in their manuscript fully available?

Reviewer #1: No

Reviewer #2: Yes

5. Is the manuscript presented in an intelligible fashion and written in standard English?

Reviewer #1: Yes

Reviewer #2: Yes

6. Review Comments to the Author

Reviewer #1: the authors responded and addressed all the amended comments and no further comments required. Congratulation to their submission

Reviewer #2: (No Response)

7. PLOS authors have the option to publish the peer review history of their article (what does this mean?). If published, this will include your full peer review and any attached files.

Reviewer #1: **Yes: **Amr ELNEMR

Reviewer #2: No

---

## [Editor Report · Acceptance letter]

1 May 2024

PONE-D-23-40746R1 

PLOS ONE

Dear Dr. Su, 

I'm pleased to inform you that your manuscript has been deemed suitable for publication in PLOS ONE. Congratulations! Your manuscript is now being handed over to our production team.

Kind regards, 

on behalf of

Dr. Paul Awoyera 

Academic Editor

PLOS ONE